

# The MONARCH high-resolution reanalysis of desert dust aerosol over Northern Africa, the Middle East and Europe (2007-2016)

Enza Di Tomaso[1], Jerónimo Escribano[1], Sara Basart[1], Paul Ginoux[2], Francesca Macchia[1], Francesca Barnaba[3], Francesco Benincasa[1], Pierre-Antoine Bretonnière[1], Arnau Buñuel[1], Miguel Castrillo[1], Emilio Cuevas[4], Paola Formenti[5], María Gonçalves[1,6], Oriol Jorba[1], Martina Klose[1,7], Lucia Mona[8], Gilbert Montané[1], Michail Mytilinaios[8], Vincenzo Obiso[1,9], Miriam Olid[1], Nick Schutgens[10], Athanasios Votsis[11,12], Ernest Werner[13], and Carlos Pérez García-Pando[1,14]

[1]Barcelona Supercomputing Center (BSC), Barcelona, Spain
[2]Geophysical Fluid Dynamics Laboratory (GFDL), Princeton, NJ, USA
[3]Consiglio Nazionale delle Ricerche-Istituto di Scienze dell'Atmosfera e del Clima (CNR-ISAC), Italy
[4]Izaña Atmospheric Research Center, AEMET, Santa Cruz de Tenerife, Spain
[5]LISA, UMR CNRS 7583, Université Paris-Est-Créteil, Université de Paris, Institut Pierre-Simon Laplace (IPSL), Créteil, France
[6]Universitat Politècnica de Catalunya - BarcelonaTech (UPC), Department of Project and Construction Engineering, Terrassa, Spain
[7]Karlsruhe Institute of Technology (KIT), Institute of Meteorology and Climate Research (IMK-TRO), Department Troposphere Research, Karlsruhe, Germany
[8]Consiglio Nazionale delle Ricerche-Istituto di Metodologie per l'Analisi Ambientale (CNR-IMAA), Italy
[9]NASA Goddard Institute for Space Studies (GISS), New York, NY, USA
[10]Department of Earth Science, Vrije Universiteit Amsterdam, 1081 HV Amsterdam, the Netherlands
[11]University of Twente, Department of Governance and Technology for Sustainability (BMS-CSTM), the Netherlands
[12]Finnish Meteorological Institute (FMI), Weather and Climate Change Impact Research, Finland
[13]State Meteorological Agency (AEMET), Spain
[14]ICREA, Catalan Institution for Research and Advanced Studies, Barcelona, Spain

**Correspondence:** Enza Di Tomaso (enza.ditomaso@bsc.es)

**Abstract.** One of the challenges in studying desert dust aerosol along with its numerous interactions and impacts is the paucity of direct in-situ measurements, particularly in the areas most affected by dust storms. Satellites typically provide column-integrated aerosol measurements, but observationally-constrained continuous 3D dust fields are needed to assess dust variability, climate effects and impacts upon a variety of socio-economic sectors. Here, we present a high resolution regional

5    reanalysis data set of desert dust aerosols that covers Northern Africa, the Middle East and Europe along with the Mediterranean sea and parts of Central Asia, and the Atlantic and Indian Oceans between 2007 and 2016. The horizontal resolution is 0.1° latitude × 0.1° longitude, and the temporal resolution is 3 hours. The reanalysis was produced using Local Ensemble Transform Kalman Filter (LETKF) data assimilation in the Multiscale Online Non-hydrostatic AtmospheRe CHemistry model (MONARCH) developed at the Barcelona Supercomputing Center (BSC). The assimilated data are coarse-mode dust

10   optical depth retrieved from the Moderate Resolution Imaging Spectroradiometer (MODIS) Deep Blue Level 2 products. The reanalysis data set consists of upper air (dust mass concentrations and extinction coefficient), surface (dust deposition and solar irradiance fields, among them) and total column (e.g., dust optical depth and load) variables. Some dust variables, such





as concentrations and wet and dry deposition, are expressed for a binned size distribution that ranges from 0.2 to 20 μm in particle diameter. Both analysis and first-guess (analysis-initialized simulation) fields are available for the variables that are diagnosed from the state vector. A set of ensemble statistics is archived for each output variable, namely the ensemble mean, standard deviation, maximum and median. The spatial and temporal distribution of the dust fields follows well-known dust cycle features controlled by seasonal changes in meteorology and vegetation cover. The analysis is statistically closer to the assimilated retrievals than the first-guess, which proves the consistency of the data assimilation method. Independent evaluation using AERONET dust-filtered optical depth retrievals indicates that the reanalysis data set is highly accurate (mean bias=-0.05, RMSE=0.12, r=0.81 when compared to retrievals from the spectral de-convolution algorithm on a 3-hourly basis). Verification statistics are broadly homogeneous in space and time with regional differences that can be partly attributed to model limitations (e.g., poor representation of small-scale emission processes), presence of aerosols other than dust in the observations used in the evaluation, and differences in the number of observations among seasons. Such a reliable high-resolution historical record of atmospheric desert dust will allow a better quantification of dust impacts upon key sectors of society and economy, including health, solar energy production and transportation. The reanalysis data set (Di Tomaso et al., 2021) is distributed via a Thematic Real-time Environmental Distributed Data Service (THREDDS) at BSC and freely available at http://hdl.handle.net/21.12146/c6d4a608-5de3-47f6-a004-67cb1d498d98.

## 1 Introduction

Desert (or mineral) dust is the most abundant aerosol by mass in the global atmosphere (Textor et al., 2006) and plays a key role in the Earth system (Knippertz and Stuut, 2014). It is emitted from the surface by aeolian processes and is originated predominantly from - but not limited to - desert regions. Dust affects weather and climate by perturbing the radiative balance directly through scattering and absorption of solar and thermal radiation (Pérez at al., 2006; Boucher et al., 2019; Miller et al., 2014), and indirectly by altering cloud formation and cloud chemistry (Cziczo et al., 2013; Harris et al., 2013; Kiselev et al., 2017). It also contributes to the fertilisation of the ocean (Jickels et al., 2005; Kanakidou et al., 2018) and the land (Yu et al., 2015; Rizzolo et al., 2017) through the deposition of iron and phosphorus, thus affecting the global carbon cycle. All in all, the amount, spatial distribution and variability of desert dust have implications on climate, the environment, air quality (Rodríguez et al., 2001; Pey at al., 2013; Barnaba et al., 2017) and human health (Mallone et al., 2011; Morman and Plumlee, 2013; Pérez García-Pando et al., 2014; Pandolfi at al., 2014; Terradellas et al., 2015; Stafoggia et al., 2016; Querol et al., 2019), and a variety of socio-economic sectors such as aviation and solar energy production (Schroedter-Homscheidt et al., 2012; Votsis et al., 2020). Due to the nature of its emission and transport, and its relatively short lifetime (Gließ et al., 2021), dust varies strongly in space and time, which requires continuous monitoring both in-situ and remotely by satellite, airborne and ground-based sensors (Barnaba and Gobbi, 2004; Kaufman et al., 2005; Marticorena et al., 2010; Kim et al., 2011; Mona et al., 2012; Pey at al., 2013; Luo et al., 2015). A major challenge in studying desert dust along with its impacts is the paucity of direct in-situ measurements in the regions most affected by dust storms. There are some operational visibility observations providing qualitative estimates of dust presence (Mahowald et al., 2007), but there is a severe lack of routine surface aerosol concentration measurements



(Benedetti et al., 2018). In addition to the lack of in-situ observations, there is limited information on aerosol speciation, which is essential to distinguish dust from other aerosol types (Rodríguez et al., 2012). Satellites mostly provide column-integrated aerosol information, but spatially and temporally-resolved surface dust concentration and deposition estimates are needed to enable detailed impact assessments. Dust observations or retrievals are therefore best exploited in combination with model
simulations either to provide optimal initial conditions (analyses) to forecast models (Benedetti et al., 2014) or to monitor current and past states of the atmosphere through the production of reanalyses, i.e., complete and consistent four-dimensional reconstructions of the atmosphere.

There are several available global aerosol reanalyses including desert dust. These include MERRA-2 (Modern-Era Retrospective analysis for Research and Applications, Version 2; Gelaro et al., 2017; Randles et al., 2017; Buchard et al., 2017)
and CAMSRA (Copernicus Atmosphere Monitoring Service Reanalysis; Inness et al., 2019) along with their predecessors MERRAero (Modern Era Retrospective analysis for Research and Applications Aerosol Reanalysis; Buchard et al., 2015) and MACC-II (Monitoring Atmospheric Composition and Climate -II, Inness et al., 2013; Cuevas et al., 2015), respectively; and the JRAero (Japanese Reanalysis for Aerosol; Yumimoto et al., 2017) and the NAAPS (Naval Research Laboratory Aerosol Analysis and Prediction System; Lynch et al., 2016) reanalyses. These global data sets have been produced at relatively coarse spatial
resolution and by assimilating total aerosol optical depth (AOD). MERRA-2 is NASA's latest reanalysis. It has been produced at a spatial resolution of 0.58° latitude × 0.6258° longitude, with 72 hybrid-eta layers, and by assimilating bias-corrected, neural network-retrieved AOD from the Moderate Resolution Imaging Spectroradiometer (MODIS) and from the Advanced Very High Resolution Radiometer (AVHRR; over ocean only), as well as AOD from the Multiangle Imaging SpectroRadiometer (MISR; over bright surfaces only) and from the Aerosol Robotic Network (AERONET) of Sun photometers. The latest reanal-
ysis for atmospheric composition produced by the Copernicus Atmosphere Monitoring Service (CAMS) CAMSRA covers the period January 2003 to 2020 and is extended by adding 1 year each year. It has been produced at a spatial resolution of ~80 km and with 60 hybrid sigma/pressure levels in the vertical, and by assimilating Collection 6 MODIS AOD produced with Deep Blue (DB; over land) and Dark Target (over land and ocean) algorithms, and by additionally assimilating the Advanced Along-Track Scanning Radiometer (AATSR) AOD from 2003 to March 2012. JRAero is a global 5-year (2011–2015) reanal-
ysis product constructed by the Meteorological Research Institute of the Japan Meteorological Agency. It has been produced assimilating MODIS 6-hourly Level 3 AOD product provided by the US Naval Research Laboratory (NRL) and the University of North Dakota (UND) for the purpose of aerosol data assimilation and which is based on the NASA operational MODIS Level 2 Collection 5 (Dark Target) AOD dataset. This same data set has been previously used, together with MISR AOD, by NRL to produce the NAAPS 11-year (2003–2013) global gridded aerosol reanalysis product at a resolution of 1° latitude ×
1° longitude.

At European level, air quality regional reanalyses (including dust) are produced by nine different operational systems and the associated multi-model ensemble through the CAMS regional services of the Copernicus programme. These models assimilate surface observations of $O_3$, $SO_2$, $NO_2$ and $CO$, and particulate matter ($PM_{2.5}$ and $PM_{10}$) operationally, and one of the models additional assimilates AOD in research mode. These products are restricted to an extended European domain, which
excludes major desert dust sources in Northern Africa and the Middle East. These reanalyses are produced as an improved





product over the daily CAMS analyses, by using the latest validated observations, but we note they may not be consistent over the different production periods, as they are not necessarily produced with the same model version.

We present here a regional reanalysis focusing specifically on desert dust aerosols that overcomes some of the potential limitations of existing global and regional reanalysis products. The data set was obtained by combining satellite remote sensing dust retrievals with a dynamical model. It spans a 10-year period, from 2007 to 2016, has a horizontal resolution of 0.1° latitude × 0.1° longitude, and 3-hourly output. It provides a regional reconstruction of past dust conditions across Northern Africa, the Middle East and Europe, and including the Mediterranean sea and parts of Central Asia, and the Atlantic and Indian Oceans. The reanalysis consists of a set of dust geophysical variables (and their uncertainties) produced with a consistent model and data assimilation scheme, i.e. a frozen version of the code used during the whole simulation period, including emission schemes, input data sets and retrieval algorithm for the assimilated observations. This ensures the production of a consistent data set avoiding the introduction of spurious trends that could be associated to model or assimilation changes.

We have adopted an ensemble-based data assimilation scheme for the estimation of the dust analysis. The use of ensemble model simulations has allowed for the estimation of flow-dependent background uncertainty which is otherwise difficult to estimate due to the highly varying nature of dust concentrations. Assimilating AOD may not necessarily constrain individual aerosol components because the aerosol attribution in the analysis increments are typically determined by the model first-guess (Tsikerdekis et al., 2021). To at least partly overcome this limitation, we have directly assimilated dust retrievals, namely satellite-derived coarse-mode dust optical depth ($DOD_{coarse}$) at 550 nm over land surfaces, including bright surfaces such as desert areas. The assimilated retrievals are based on the MODIS DB algorithm (Hsu el al., 2013; Sayer et al., 2013), which uses measurements at different wavelengths with a different contrast between the surface and atmospheric aerosols. In particular, the algorithm capitalizes on the much lower surface reflectance at ultraviolet wavelengths than at longer wavelengths.

This new reanalysis data set can be used to support the provision of climate services and monitoring. It can also contribute to the development of dust impact mitigation strategies. For instance, the design of the reanalysis output fields has been tailored to the specific needs in three socio-economic sectors affected by mineral dust, which are air quality and health, energy production and transport. In addition to the 3D fields of dust mass concentration, the reanalysis data set includes dust extinction and deposition variables, along with other variables associated with meteorology and radiation. In summary, we present here a regional dust reanalysis at an unprecedented resolution using for the first time specific dust retrievals over dust source regions and including grid-level uncertainty estimates.

The following sections describe the different aspects related to the production of the reanalysis: the dust modeling aspect, including the dust sources and emission schemes outlined in Sect. 2; the generation of ensemble perturbations to best characterize model uncertainty explained in Sect. 3; the assimilated dust retrievals and the data assimilation scheme described, respectively, in Sect. 4 and 5. Additionally, Sect. 6 describes the details of the reanalysis simulation settings, while Sect. 7 describes the content and structure of the reanalysis data set. Section 8 provides an evaluation of the column-integrated dust optical depth (DOD) and $DOD_{coarse}$ in terms of geographical distribution, study of analysis increments, data assimilation inner diagnostics, and comparison against independent observations. Information about the data set availability is provided in Sec. 9. Finally, conclusions are drawn in Sect. 10.



## 2 MONARCH modeling system

The reanalysis has been produced using the Multiscale Online Non-hydrostatic AtmospheRe CHemistry model (MONARCH; Pérez et al., 2011; Haustein et al., 2012; Jorba et al., 2012; Spada et al., 2013; Badia et al., 2017; Klose et al., 2021), which consists of advanced chemistry and aerosol packages coupled online with the Non-hydrostatic Multiscale Model on the B-grid

(NMMB; Janjic et al., 2001; Janjic and Gall, 2012). MONARCH is able to work across a wide range of spatial scales thanks to its unified non-hydrostatic dynamical core. In the global setup, MONARCH is run on a latitude-longitude grid, while the regional version used in this work runs on a rotated latitude-longitude grid. Different physics schemes are available in the NMMB to resolve turbulence, convection, soil, radiation and clouds. The exact configuration used in this work is reported in Table 1 where the key configuration settings are summarized for both modelling and data assimilation aspects.

MONARCH represents the atmospheric dust cycle including emission, transport and deposition along with dust-radiation interactions. A variety of dust emission schemes and configurations are available as described in Klose et al. (2021), ranging from strongly simplified to physics-based parameterizations. Dust transport is produced by horizontal advection, solved with the Adams–Bashforth scheme, vertical advection, solved with the Crank–Nicholson scheme, and lateral diffusion, which follows the Smagorinsky non-linear approach. Furthermore, dust is vertically mixed by turbulent diffusion and deep and shallow

convection. Sinks include gravitational settling, dry deposition through turbulent diffusion, and in-cloud and below-cloud scavenging from both stratiform and convective clouds. MONARCH follows a sectional approach for dust, i.e. the size distribution is decomposed into small size bins that range from 0.2 to 20 µm in diameter. The particle-size distribution (PSD) at emission can either be chosen from a set of pre-defined PSDs or is calculated online, depending on the selected emission scheme. In this work, we have used a PSD of emitted dust over sources derived from Kok (2011).

A more detailed description of the dust module of MONARCH can be found in Pérez et al. (2011) and Klose et al. (2021), with the latter work including also advances developed after the start of the dust reanalysis production. Those recent developments were therefore not yet used in the present work for which a frozen model version is important. Below we provide further details on the configuration of the emission and radiation schemes used in this work.

### 2.1 Dust emission schemes

MONARCH contains multiple dust emission schemes, of which we used the following three to generate ensemble perturbations for the production of the reanalysis: (i) a scheme based on Marticorena and Bergametti (1995), hereafter called MB95, which is based on saltation flux and soil texture and was combined with the topographic source mask from Ginoux et al. (2001) as described in Pérez et al. (2011); (ii) the GOCART dust emission scheme from Ginoux et al. (2001) based mainly on a topographic source function, hereafter called G01; (ii) a scheme based on brittle fragmentation by saltation as in Kok et al.

(2014), hereafter called K14. The location of dust sources is identified by a climatology of frequency of occurrence (FoO) of DOD greater than 0.2 derived from the MODIS DB Collection 6 at the resolution of 0.1° latitude × 0.1° longitude (Hsu et al., 2004; Ginoux et al., 2012, see Sec. 4.3.1) with a minimal threshold for FoO equal to 0.05, below which there is no emission. Surface roughness is accounted for in the dust emission calculation using the drag partition parameterization from Marticorena





**Table 1.** Overview of the characteristics of the reanalysis.

|  | Reanalysis configuration |
| --- | --- |
| **Domain, resolution and output** |  |
| data set length | 10 years (2007-1016) |
| output frequency | 3 hours (starting at 3 UTC) |
| geographical domain | regional |
| horizontal resolution | 0.1° latitude × 0.1° longitude |
| vertical resolution | 40 hybrid pressure-sigma layers / 15 pressure levels (1000-100 hPa) |
| top pressure | 50 hPa |
| output variables | 6 (surface), 3 (total column), 3 (upper air) |
| uncertainty estimation | based on the spread in the MONARCH ensemble (12 members) |
| **Data assimilation (DA)** |  |
| assimilation algorithm | ensemble-based DA (4D-LETKF; Hunt et al., 2007; Schutgens et al., 2010; Di Tomaso et al., 2017) |
| control vector | 3D mixing ratio of dust coarse bins (ranging from 1.2 to 20 µm in dust particle diameter) |
| assimilated observations | MODIS DB $DOD_{coarse}$ at 550 nm (Ginoux et al., 2010, 2012; Pu and Ginoux, 2016) |
| observation satellite platform | NASA Aqua (EOS PM-1) |
| observational coverage | clear-sky, snow-free, land and day-time |
| **Chemical weather system** |  |
| aerosol model | MONARCH (Multiscale Online Non-hydrostatic AtmospheRe CHemistry model v1.0, with improvements; Pérez et al., 2011; Klose et al., 2021) |
| dust emission scheme | MB95 (Marticorena and Bergametti, 1995), G01 (Ginoux et al., 2001), K14 (Kok et al., 2014) |
| particle-size distribution at emission (before perturbation) | PSD as in Kok (2011) |
| meteorological model | NMMB (Non-hydrostatic Multi-scale Model on the B grid; Janjic and Gall, 2012) |
| meteorological initialization | ERA-interim (Dee el al., 2011) and MERRA-2 (Gelaro et al., 2017) with ERA-5 soil information (Hersbach et al., 2020) |
| radiation scheme | RRTM (Iacono et al., 2008) LW: OPAC RIs (Hess et al., 1998); SW: mineralogy-based RIs (Gonçalves et al., 2021, in prep.) spherical particle shape |
| microphysics scheme | Ferrier (Ferrier et al., 2002) |
| surface layer | NMMB similarity theory (Janjic, 1994, 1996b) |
| land surface scheme | Noah (Ek et al., 2003) |
| turbulence scheme | Mellor-Yamada-Janjic (Janjic, 1996a, 2002) |
| convection scheme | Betts-Miller-Janjic (Betts, 1986; Betts and Miller, 1986; Janjic, 1994, 2000) |
| ensemble generation | multi-parameter, multi-physics source perturbations and multi-meteorological initial and boundary conditions |



and Bergametti (1995) with input from MODIS Collection 5 monthly Leaf Area Index for the specific year of simulation from
2007 to 2015 and from a climatology for 2016, combined with a static roughness length for arid regions (Prigent et al., 2012)
as described in Klose et al. (2021). The X-parameter in the Marticorena and Bergametti (1995) drag partition follows Pierre
et al. (2014). The USGS climatological database for vegetation is used by the meteorology and land-surface scheme. A soil
moisture correction is used for MB95 and K14 as in Fecan et al. (1999) with a revised scaling factor as in Klose et al. (2021)
and Zender et al. (2003). G01 uses the default GOCART soil moisture correction, which is based on Belly et al. (1964) as
described in Ginoux et al. (2001), and a threshold friction velocity as described in Pérez et al. (2011).

## 2.2 Radiation and dust optical properties

In MONARCH, dust is coupled online with the RRTMG radiation scheme, which accounts for shortwave (SW) absorption and
scattering and longwave (LW) absorption (Iacono et al., 2008). The input dust optical properties (extinction efficiency, single
scattering albedo and asymmetry factor) for each particle size bin and wavelength are based on refractive indices (RIs) that
account for the variation in mineralogical composition by size (Perlwitz et al., 2015a,b; Scanza et al., 2015; Pérez García-Pando
et al., 2016) in the SW and derived from the OPAC dataset (Hess et al., 1998) the LW. Optical properties are calculated using
Mie scattering theory (Mishchenko et al., 2002) assuming that dust is spherical despite its well-known non-sphericity (Kok et
al., 2017). Although MONARCH now allows to account for the effect of dust non-sphericity upon the optical properties (Klose
et al., 2021), this option was not ready by the start of the reanalysis production.
To calculate the mineralogy-based size-dependent RIs in the SW, we applied the multi-component Maxwell Garnett theory
(Markel, 2016) to internal mixtures of eight dominant dust minerals (Gonçalves et al., 2021, in prep.) derived from the soil
mineralogical atlas of Claquin et al. (1999). The single-mineral RIs were taken from Scanza et al. (2015). The mineral fractions
in each size bin are estimated for each of the 28 soil types considered in the atlas based on brittle fragmentation theory (Kok,
2011). For each size bin and wavelength, we finally retain the median real and imaginary RIs across the 28 soil types. In the
visible band, the obtained median RIs compare well with recent chamber-based retrievals (Di Biagio et al., 2019) and in-situ
aircraft measurements (Denjean et al., 2016), as shown in Gonçalves et al. (2021, in prep.).

The dust-radiation coupling allows the computation of the direct radiative effect at each radiation time step with a simple
double call approach. We also calculate direct normal irradiance (DNI) and global horizontal irradiance (GHI) at the surface,
under all-sky conditions, from downward fluxes in ultraviolet-visible-near infrared bands of the model. While GHI includes
direct and diffuse beams collected by a horizontal unit surface, DNI accounts for the direct beam hitting a normal surface.
These variables are useful for applications in the context of solar energy production.

## 3 Generation of ensemble perturbations

We adopted an ensemble-based data assimilation scheme to estimate dust. Hence model uncertainty, expressed as background
error covariance in the data assimilation algorithm, is estimated from the realizations of the dust fields in an ensemble of
MONARCH model calculations. The use of an ensemble of model simulations allows the estimation of a flow-dependent





background uncertainty that would otherwise be difficult to estimate due to the highly variable nature of dust concentrations. We generated a 12-member ensemble using different meteorological initial and boundary conditions, and dust emission schemes, along with additional perturbations in the model emission parameters. Such perturbations aim at representing the model uncertainty, mainly in dust emission, which is one of the major contributors to model error (Huneeus et al., 2011). The characteristics of each ensemble member are listed in Table S1 and described below.

The benefit of combining meteorological and aerosol source perturbations was showed in Rubin et al. (2016). The meteorology in our reanalysis is re-initialized everyday using global reanalyses. Our ensemble uses two different meteorological reanalyses as initial conditions at the start of every daily run (at 0 UTC), and as boundary conditions every 6 hours. ERA-Interim (Berrisford et al., 2011; Dee el al., 2011) is used in 6 ensemble members and MERRA-2 (Gelaro et al., 2017) together with ERA5 soil information (Hersbach et al., 2020) is used in the remaining 6 members.

Experiments conducted in Escribano et al. (2021) showed that using different dust emission schemes provides a better characterization of the background covariance than a single scheme with parameter perturbations, due to the large variability in the modelled emissions. The ensemble uses three different emission schemes briefly introduced in Sect. 2, namely MB95 (as in Pérez et al., 2011), G01 (as in Ginoux et al., 2001) and K14 (as in Kok et al., 2014). Each emission scheme was used four times (twice in each of the two 6-member groups driven by the different meteorological reanalyses). In addition, each of the 12 ensemble members was run with a different value for one or more parameters in the corresponding emission scheme following Di Tomaso et al. (2017). Specifically, we perturbed the threshold friction or wind velocity, which is soil moisture-dependent and determines the friction or wind velocity above which soil particles begin to move in saltation, and the dust emission flux across each of the eight dust model bins. The threshold friction or wind velocity was perturbed by drawing a multiplicative random factor from a normal distribution with mean 1 and spread 0.4. The dust emission flux was perturbed imposing a physical constraint. Correlated noise was used across the bins so that noise correlation decreases with increased difference in the normalized cubic radius between the bins; the noise has mean 1 and a standard deviation of 30% of the unperturbed value in each bin. These emitted size distribution perturbations used here are analogous to those in Fig. 1 in Di Tomaso et al. (2017) but departing from Kok (2011) instead of D'Almeida (1987).

## 4 Assimilated observations

We have used for assimilation an innovative DOD data set derived from the MODIS DB aerosol products (Collection 6), which covers all cloud-free and snow-free land surfaces. DB aerosol retrievals are available over areas not easily covered by other observational data sets, e.g., very bright reflective surfaces such as deserts, and are therefore particularly relevant for dust applications. The MODIS Dark Target product, for example, has a limited coverage over land since the retrieval algorithm assumes low surface albedo. The DB algorithm uses top-of-the-atmosphere reflectances at 412 and 470 nm, and, in the presence of heavy dust load, also at 650 nm. It exploits the fact that, over most surfaces, a darker surface and stronger aerosol signal is seen in the blue wavelength range than at longer wavelengths. The quality of the MODIS DB AOD product is improved in Collection 6 compared to previous releases, as shown by the work of Sayer et al. (2014) and Gkikas et al. (2015), based



respectively on Level 2 and Level 3 retrievals. Furthermore, a recent study by Schutgens et al. (2020) showed that DB AOD
from MODIS (on-board the Aqua satellite) is one of the best products when compared to other satellite products.

More specifically, we have assimilated $DOD_{coarse}$ retrieved from MODIS DB Level 2 aerosol products as described in Ginoux
et al. (2010, 2012) and Pu and Ginoux (2016). The generation of the dust retrievals includes the different steps of formatting,
dust filtering and retrieval. First, aerosol products such as AOD, single scattering albedo, and the Ångström exponent are
interpolated to a regular grid of $0.1°$ latitude $\times$ $0.1°$ longitude using the algorithm described by Ginoux et al. (2010). The
DOD is then derived from AOD following the methods of Ginoux et al. (2012) with adaptions to MODIS Collection 6 aerosol
products. To separate dust from other aerosols, two variables are used: the Ångström exponent, which is highly sensitive to
particle size (Ångström, 1929; Eck et al., 1999), and a single scattering albedo at 412 nm less than 0.95 for dust due to its
absorption of solar radiation (Takemura et al., 2002). Subsequently, an empirical continuous function relating the Ångström
exponent to fine-mode AOD (Andeson et al., 2005, their Eq. 5) is applied to retrieve the dust fine-mode fraction of optical
depth.

Since the retrievals are based on visible reflectances, their availability is limited to day-time only. The MODIS instrument
is on-board two NASA polar-orbiting satellites, namely Aqua and Terra. However, we have considered for assimilation only
$DOD_{coarse}$ retrievals based on measurements from MODIS on-board the Aqua platform. The equatorial crossing local time of
the Aqua satellite is at 1:30 p.m. in an ascending orbit. In our 3-hourly discretization of the assimilation window, the assimilated
observations are associated with the time slot (or interval) centered at 12 UTC.

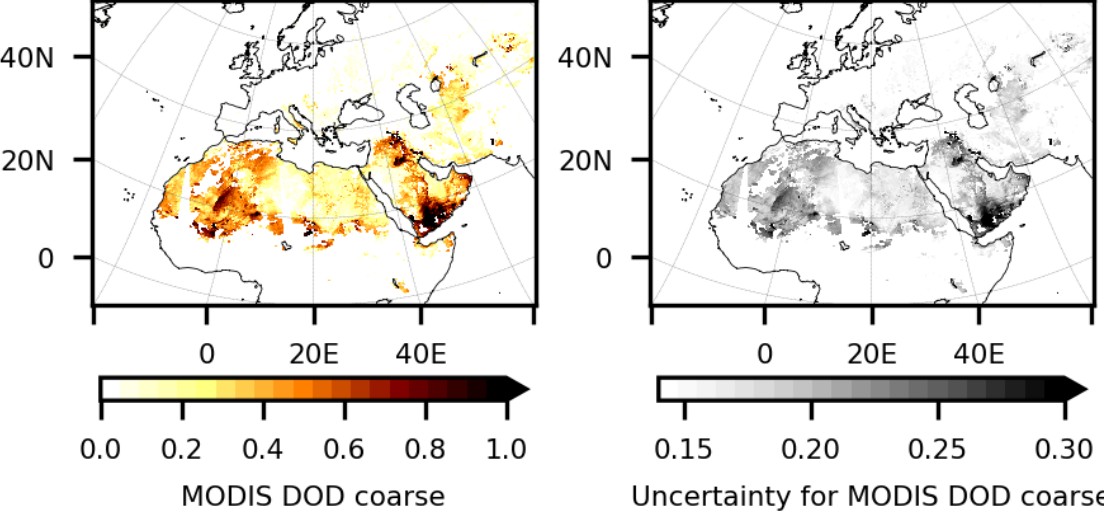

**Figure 1.** Example of assimilated observations for July 9, 2012: $DOD_{coarse}$ retrieved from the Aqua MODIS DB Level 2 products (Collection 6; left) and the associated observation uncertainty used in the assimilation algorithm (right).

We have used $0.07 + 0.075\ DOD_{coarse}$ to characterize the observation uncertainty of the assimilated observations, following
the linear model of previous studies (Hsu el al., 2013; Sayer et al., 2013) with the coefficients adjusted for our application





by inflating the uncertainty for low $DOD_{coarse}$ values, which were otherwise detrimental for the analysis. We have assumed a diagonal observation error covariance matrix, i.e. uncorrelated error between the different retrievals. Observation coordinates

were pre-processed to be mapped on the rotated longitude–latitude regional grid of MONARCH. Figure 1 shows an example of the extent of the daily observational coverage on a given date (July 9, 2012) together with the associated observational uncertainty.

Maps of observation counts are shown in Fig. 2 for the whole reanalysis period (top row of Fig. 2) and for the different seasons (row 2 to 5 of Fig. 2), namely the winter seasons represented by December, January and February (DJF), the spring

season represented by March, April and May (MAM), the summer season represented by June, July and August (JJA), and the autumn season represented by September, October and November (SON). As expected, there is a higher number of dust retrievals closer to sources than far from them. The total number of retrievals is bigger in the SON and JJA seasons than in the other seasons. During the boreal winter the number of retrievals inland from the Gulf of Guinea increases compared to other times of the year due to transport of dust by northeasterly Harmattan winds. The number of dust retrievals decreases in

the north of Europe and Asia in the DJF season as MODIS DB covers only snow-free surfaces. Yearly observation counts are consistent throughout the whole period (see Fig. S1).

## 5   Data assimilation algorithm

The reanalysis was produced using a Local Ensemble Transform Kalman Filter (LETKF) data assimilation scheme (Hunt et al., 2007; Miyoshi and Yamane, 2007; Schutgens et al., 2010; Tsikerdekis et al., 2021) coupled to the MONARCH ensemble. We

have used an implementation of the LETKF scheme with four-dimensional extension (4D-LETKF) as described in Hunt et al. (2007), in order to estimate the dust analysis over a 24-hour assimilation window. The overall scheme implements an iterative approach consisting of a forward simulation of the MONARCH ensemble for 24 hours and a state estimation step. The two steps are coupled at each iteration. The state estimation step is an execution of the LETKF which combines information from the dust observations and the model ensemble simulations. The forward simulation of the MONARCH ensemble is named

first-guess (or background) to indicate a simulation initialized from an analysis, and thus incorporates information from past observations. As a result of the estimation step, the analysis is estimated at each assimilation window using both concurrent and past observations.

The LETKF is well suited to computationally-demanding calculations such as the estimation of a high resolution analysis carried out in this work. The analysis at each model grid point can be calculated independently, and at each grid point only

observations within a certain distance are assimilated. Furthermore, the use of a dynamic characterization of model background uncertainty, through ensemble forward simulations, is well suited for highly varying dust fields. A detailed description of the scheme can be found in Hunt et al. (2007). Below we discuss the basic concepts behind the LETKF algorithm.

Consider a state vector $\boldsymbol{x}$ of the dynamic variables of a system, in our case the dust mass mixing ratio. The mean analysis increment at a grid point is estimated as a linear combination of the background ensemble perturbations $\mathbf{X}^b$ :

$$\bar{\boldsymbol{x}}^a = \bar{\boldsymbol{x}}^b + \mathbf{X}^b \mathbf{w} \tag{1}$$

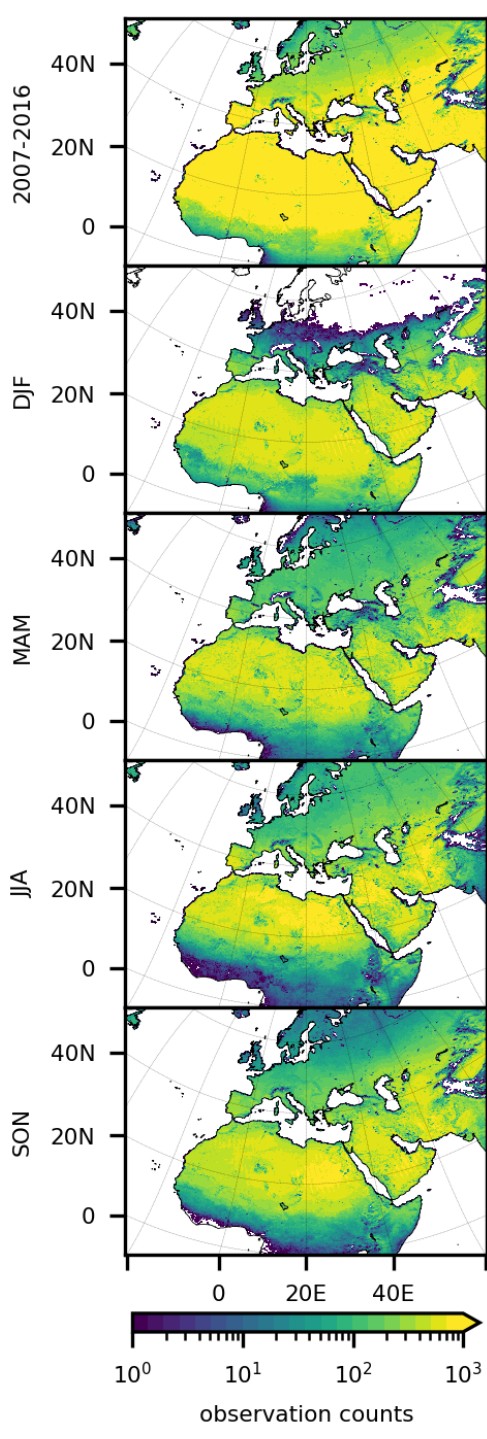

**Figure 2.** Maps of counts of assimilated observations for the whole period (2007-2016; top row) and for the different seasons (DJF, MAM, JJA, SON; rows 2 to 5) of the 10-year period.





where we use the superscripts $a$ and $b$ to denote respectively the analysis and background state vector, and where the $i$th column of the matrix $\mathbf{X}^b$ is $\boldsymbol{x}^{b(i)} - \bar{\boldsymbol{x}}^b$, $\{i = 1, 2, \ldots, k\}$ with $k$ ensemble members (12 in our case), i.e. the difference between the $i$th ensemble member $\boldsymbol{x}^{b(i)}$ and the ensemble mean $\bar{\boldsymbol{x}}^b$. $\mathbf{w}$ is termed the "weight" vector specifying what linear combination of the background ensemble perturbations is added to the background mean to obtain the analysis ensemble. The "weight" vector is given by

$$\mathbf{w} = [\mathbf{Y}^b \mathbf{R}^{-1} \mathbf{Y}^b + (k-1)\mathbf{I}]^{-1} \mathbf{Y}^b \mathbf{R}^{-1} (\boldsymbol{y}^o - \bar{\boldsymbol{y}}^b) \tag{2}$$

where $\mathbf{Y}^b$ is the background ensemble perturbation matrix in observation space (or background observation ensemble perturbation matrix), $\mathbf{R}$ is the observation error covariance matrix which we assume diagonal, $\mathbf{I}$ is the identity matrix, $\boldsymbol{y}^o$ is the vector of observations and $\bar{\boldsymbol{y}}^b$ is the mean background observation ensemble. The background observation ensemble is obtained applying the observation operator $h(\cdot)$ to the ensemble members $\boldsymbol{x}^{b(i)}$, i.e. $\boldsymbol{y}^{b(i)} = h(\boldsymbol{x}^{b(i)})$.

The 4D extension of the algorithm is coded such that background observation means $\bar{\mathbf{y}}_j$ and perturbation matrices $\mathbf{Y}_j$ are formed at the various time slots $j$ when the observations are available, then they are concatenated to form a combined background observation mean $\bar{\mathbf{y}}$ and perturbation matrix $\mathbf{Y}$, where the time slots are the time intervals into which the assimilation window is split. $\bar{\mathbf{y}}$ and $\mathbf{Y}$ are used for the calculation of a "weight" vector $\mathbf{w}$ using the standard LETKF, i.e. we calculate a single $\mathbf{w}$ based on all innovations throughout the day. This same $\mathbf{w}$ is then applied to the state vector at different times throughout the assimilation window.

Spatial covariance localization can be applied in the LETKF algorithm through R-localization, i.e. the localization is performed in the observation error covariance matrix, making the influence of an observation on the analysis decay gradually toward zero as the distance from the analysis location increases. The use of spatial localization reduces the effect of spurious long-range covariances due to sampling errors produced by a low dimensionality of the ensemble. To achieve this, the observation error is divided by a distance-dependent function that decays to zero with increasing distance: $e^{-\frac{dist^2}{l^2}}$, where $dist$ is the distance in the grid space between an observation and the model grid, and $l$ is a horizontal localization factor. The localization factor was set to 15, hence the observation influence practically fades to zero before 30 model grid points away from the observation location (in the horizontal plane).

The control variable is formulated in terms of the total mixing ratio over the 5 model prognostic variables (corresponding to different dust particle size bins) used to simulate coarse dust in MONARCH. Therefore an observation operator is needed to map the ensemble mean control vector into the observation space. The observation operator has two components: (i) a spatial interpolation of the model simulation to the observation location which is done at the observation longitude and latitude; (ii) the calculation of simulated $\mathrm{DOD_{coarse}}$ at the wavelength of $550\,\mathrm{nm}$ which is calculated using the five coarse model size bins ranging from $1.2$ to $20\,\mu\mathrm{m}$ in dust particle diameter. The analysis of the model's fine dust fraction (i.e. the three model size bins from $0.2$ to $1.2\,\mu\mathrm{m}$ in dust particle diameter) is estimated proportionally to the change (due to observation assimilation) of the coarse fraction. This choice is motivated by the fact that observations do neither carry information about fine dust particles nor about particle size distribution. Hereafter, DOD and $\mathrm{DOD_{coarse}}$ refers to the wavelength of $550\,\mathrm{nm}$.



## 6 Domain, resolution and other simulation settings

The key model and data assimilation configuration settings of the reanalysis are summarized in Table 1. The reanalysis extends over the period 2007-2016 and covers a regional domain centered around northern Africa, the Middle East and Europe (hereafter called NAMEE region), that also includes parts of Central Asia, and the Atlantic and Indian Oceans. The domain has a 0.1° latitude × 0.1° longitude horizontal resolution, and 40 hybrid pressure-sigma model layers in the vertical. The model top was set to 5000 Pa. This domain configuration is used operationally to deliver daily forecasts at the World Meteorological

Organisation Barcelona Dust Regional Center (https://dust.aemet.es/).

The model runs were conducted using a dynamics time step of 20 s. Lateral diffusion is called every time step, advection every 2 time steps, turbulence, surface layer, dust emission, sedimentation and dry deposition routines every 4 times steps, moist convection, microphysics and wet deposition every 8 time steps, and short- and longwave radiation routines were called every 180 time steps. The MONARCH ensemble of forward simulations was run daily at 0 UTC during 24 hours, which was

used as the first-guess for the data assimilation. Simulation outputs are provided every 3 hours (3:00, 6:00, 9:00, 12:00, 15:00, 18:00, 21:00 and 0:00 UTC) which is also the time resolution of the reanalysis product. Figure 3 shows the scheme of the 24-hour assimilation window for the production of the reanalysis where each ensemble member forward simulation is initialized at 0 UTC using the dust analysis produced in the previous window.

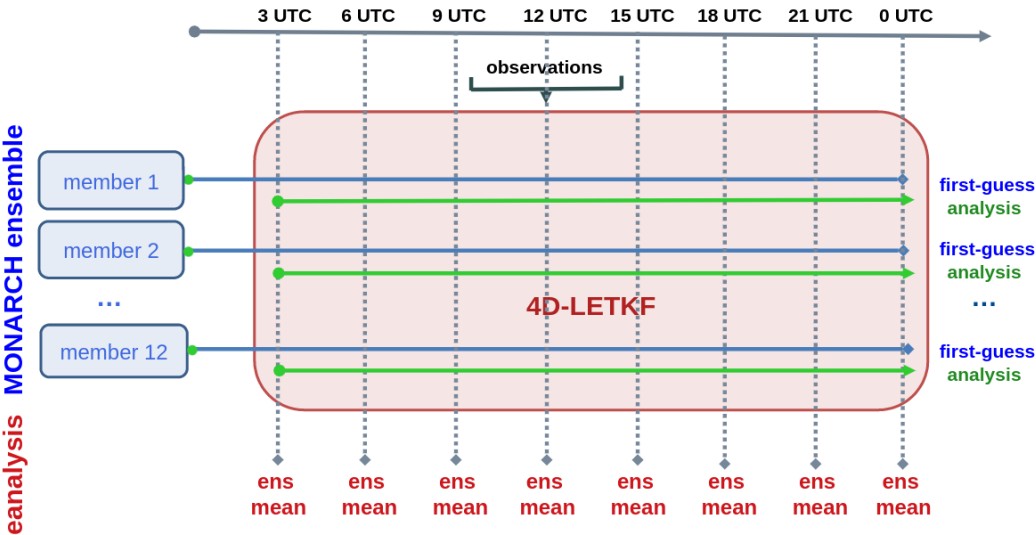

**Figure 3.** Schematic of the 24-hour assimilation window for the production of the reanalysis. The ensemble member analyses are used to initialize the corresponding ensemble member first-guess in the subsequent simulation/state estimation window.

Simulations were run without inflating the background or analysis covariance errors during the assimilation cycle. A quality

control has been applied as in Di Tomaso et al. (2017) that rejects observations by a first-guess departure check (observations





further than 1.4, in DOD$_{coarse}$, from the first-guess are rejected). This quality control is applied since the observations have not been corrected before assimilation for possible systematic biases. After the estimation of total dust coarse mixing ratio analysis, the analysis increments are partitioned among the dust coarse size bins according to their fractional contribution to the total coarse mixing ratio in the forward simulation step (i.e. before assimilation).

A spin-up period was necessary for the soil variables that need a longer period to adjust. We have run a one year spin-up with a two-member experiment, each of them initialized using either MERRA2 or ERA-Interim meteorology with ERA5 soil information. Furthermore, a two month spin-up period was needed for the ensemble without data assimilation, to have a good representation of the ensemble spread everywhere in the atmospheric domain.

### 6.1    Ensemble calibration

MONARCH uses a globally uniform, empirically constrained, tuning (or calibration) factor for the total emitted dust mass, referred to as $d_{cal}$. This factor varies according to the specified configuration settings for the simulation. In particular, it depends on the emission scheme, and the meteorological initial and boundary conditions used to initialize the simulation. We calibrated 6 free running experiments, which cover all the different combinations between emission scheme and meteorological conditions. The calibration factors were obtained by rescaling initial values for the calibration factors, namely $d_{cal}(m)_{old}$, by

the ratio between the MODIS DB mean DOD$_{coarse}$ and the ensemble free-run mean DOD$_{coarse}$ calculated over the whole domain, i.e.:

$$d_{cal}(m)_{new} = d_{cal}(m)_{old} \frac{DOD_{coarse,MODIS}}{DOD_{coarse,model}} \text{ with } m = 1,...,6 \qquad (3)$$

where $m$ indicates an ensemble member. We have repeated the estimation twice where the second simulation re-run has used the calibration factors estimated from the first run. The final estimated calibration factors for each of 6 ensemble members are

reported in the Table S2.

### 7    Reanalysis product description

The reanalysis data set consists of three-dimensional (3D) and two-dimensional (2D) variables (see Table 2). The 3D, or upper air, variables include dust mass concentration [kg m$^{-3}$] for each dust size bin, dust extinction coefficient at 550 nm [m$^{-1}$] integrated over all size bins, and height [m]. The 2D variables are either surface fields or total column fields. The 2D

variables for each dust size bin include accumulated dust dry and wet deposition over the previous 3 hours [kg m$^{-2}$ 3 h$^{-1}$], and instantaneous total column dust load [kg m$^{-2}$], dust mass surface concentration [kg m$^{-3}$], DOD [unitless] and DOD$_{coarse}$ [unitless] at 550 nm. The set of archived 2D variables is completed by the surface extinction coefficient at 550 nm [m$^{-1}$], direct normal irradiance [W m$^{-2}$], and global horizontal irradiance [W m$^{-2}$]. These variables have been used to produce dust-relevant information for different sectors (Votsis et al., 2020, 2021) and related validation exercises (Mytilinaios et al., 2021,

2022, in prep.). For example, a dust-PM10 field has been derived from the 2D, bin-resolved dust mass surface concentration for air-quality applications, visibility data from 3D dust-extinction coefficient fields have been used for aviation applications



(Basart et al., 2021), while soiling index based on wet and dry dust deposition has been used to develop products for solar energy production (Rautio et al., 2021, in prep.).

**Table 2.** List of reanalysis variables. For each variable the following ensemble statistics are calculated and archived: ensemble mean, standard deviation, max and median.

| variable description (name in archive) | unit | spatial dimension | description of dust particle size | first-guess | analysis |
|---|---|---|---|---|---|
| dust concentration (concdubin1-8) | $kg\,m^{-3}$ | 3D | 8 bins | ✓ | ✓ |
| direct normal irradiance (dni) | $W\,m^{-2}$ | 2D | NA | ✓ | |
| accumulated dry deposition over the previous 3 hours (drydu) | $kg\,m^{-2}\,3\,h^{-1}$ | 2D | 8 bins | ✓ | |
| dust extinction coefficient at 550 nm (ec550du) | $m^{-1}$ | 3D | total | ✓ | ✓ |
| global horizontal irradiance (ghi) | $W\,m^{-2}$ | 2D | NA | ✓ | |
| dust load (loaddu) | $kg\,m^{-2}$ | 2D | 8 bins | ✓ | ✓ |
| dust optical depth at 550 nm (od550du) | unitless | 2D | total | ✓ | ✓ |
| coarse dust optical depth at 550 nm (od550ducoarse) | unitless | 2D | total | ✓ | ✓ |
| dust surface concentration (sconcdubin1-8) | $kg\,m^{-3}$ | 2D | 8 bins | ✓ | ✓ |
| dust surface extinction coefficient (sec550du) | $m^{-1}$ | 2D | total | ✓ | ✓ |
| accumulated wet deposition over the previous 3 hours (wetdu) | $kg\,m^{-2}\,3\,h^{-1}$ | 2D | 8 bins | ✓ | |
| height (z) | $m$ | 3D | NA | ✓ | |

Both analysis and first-guess fields are available for the variables that are diagnosed from the state vector. As mentioned earlier, the first-guesses are model forward simulations initialized with an analysis. When available, the analysis field is the recommended output for that variable. A set of ensemble statistics is calculated and archived for each output variable, namely the ensemble mean, standard deviation, maximum and median. The spread among the ensemble members, represented by the standard deviation with respect to the ensemble mean, can be interpreted as a measure for the uncertainty in the mean estimates. Figure 4 shows the ensemble mean over the whole reanalysis period for the analysis or first-guess of some of the 2D variables. While model fields have been produced on 40 vertical levels, the data are stored on 15 standard pressure levels between 1000 and 100 hPa (i.e., 1000, 975, 900, 850, 750, 700, 600, 500, 400, 350, 300, 250, 175, 150, 100 hPa), which were defined taking into account regulatory standards in the aviation sector (in view of end-user products developed from the reanalysis in this sector; Votsis et al., 2020). In that way we reduced storage space while easing the use of the vertical information.

The reanalysis data set is structured into individual Network Common Data Format (NetCDF) files per variable and type of ensemble statistics. Further details on the file structure of the data set are reported in Sec. 9, while the naming convention for the data set files and folders is explained in Appendix A.

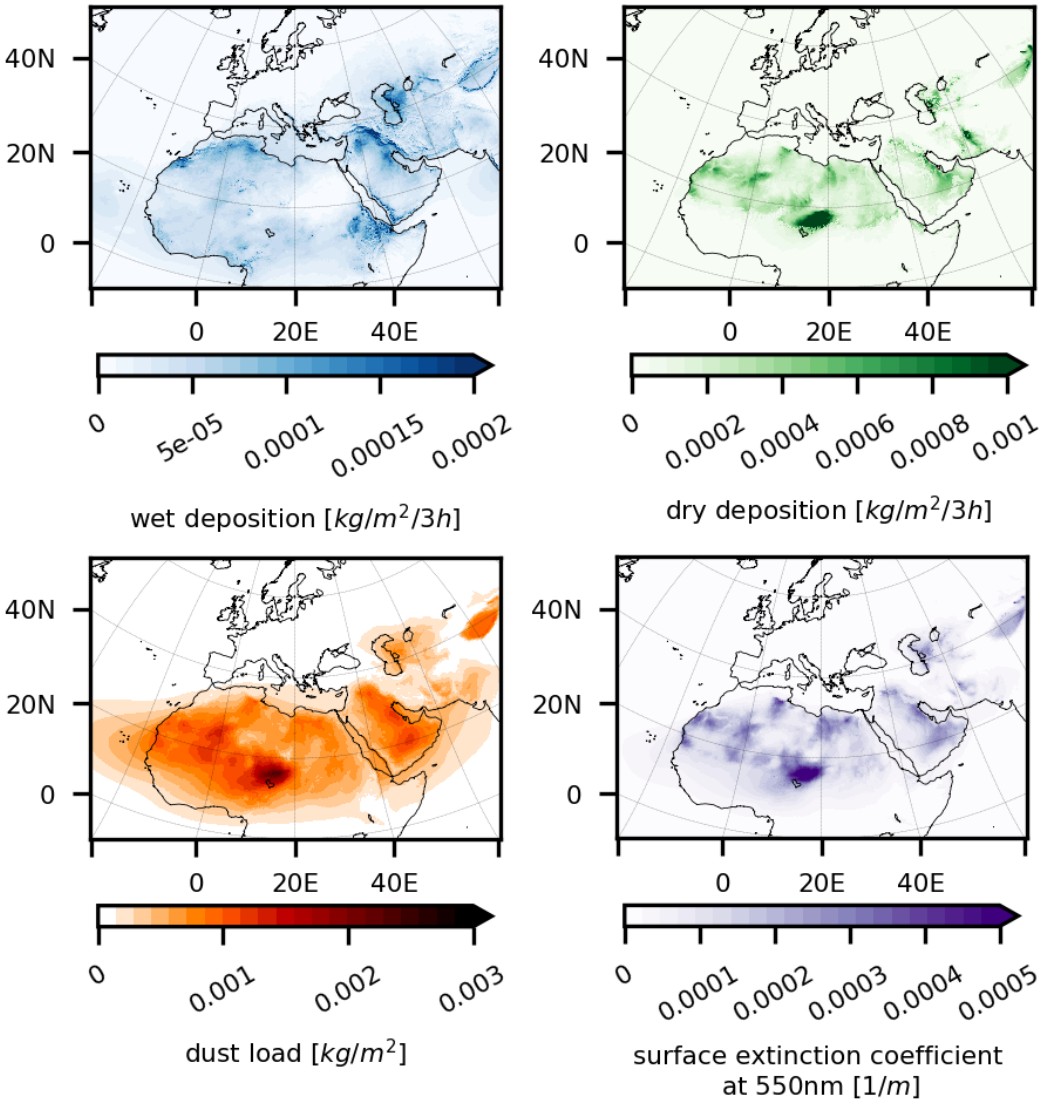

**Figure 4.** Maps of mean 3-hourly first-guess of accumulated (over the previous 3 hours) dust wet and dry deposition $[\mathrm{kg\,m^{-2}\,3\,h^{-1}}]$, analysis of total column dust load $[\mathrm{kg\,m^{-2}}]$ and of dust surface extinction coefficient at 550 nm $[\mathrm{m^{-1}}]$ calculated for the whole period (2007-2016). Model fields are ensemble mean.





## 8 DOD evaluation

In this section we validate the reanalysis DOD or DOD_coarse in terms of data assimilation inner diagnostics (analysis increments and statistics of departures from assimilated observations) and verify it against independent ground-based observations. We

also discuss the DOD spatial and temporal patterns over the reanalysis domain and period. Figure 5 highlights the location of major dust source areas that will be used in the discussion. The verification of DOD and DOD_coarse against long-term ground-based observations across the domain is a first step towards a more comprehensive evaluation of the reanalysis data set that is planned in follow-up papers (Mytilinaios et al., 2021, 2022).

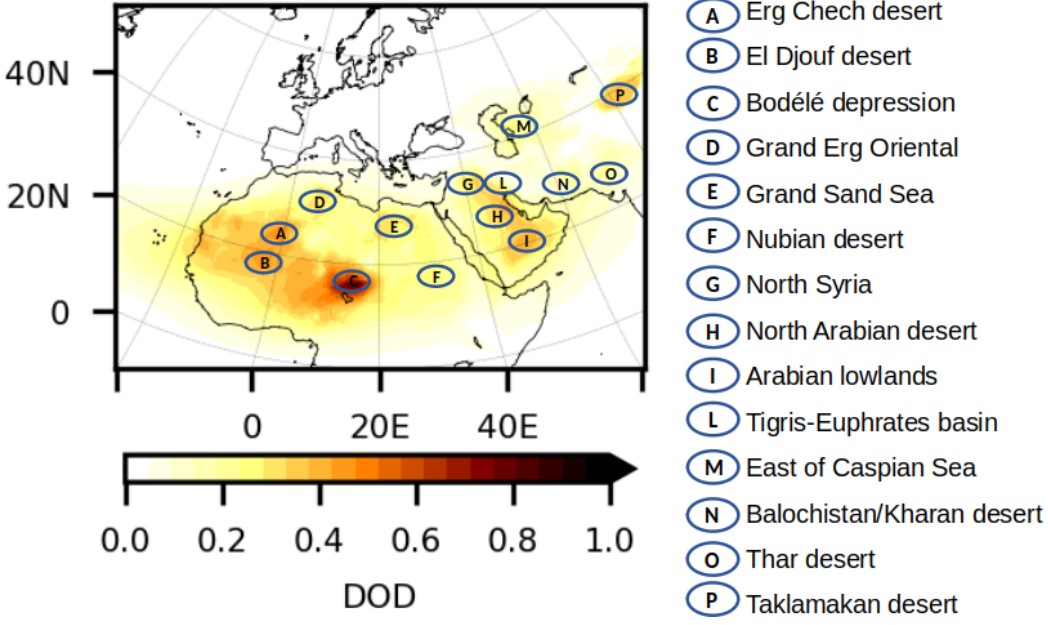

**Figure 5.** A number of desert, arid and semi-arid regions of interest for the description of the dust reanalysis. The underlying dust field is the mean 3-hourly DOD analysis calculated for the whole reanalysis period (2007-2016).

### 8.1 DOD geographical distribution

Figure 6 shows the ensemble annual and seasonal mean DOD for the first-guess (left column) and the analysis (central column) during the whole reanalysis period. In agreement with observations, the highest DOD values are placed over the major emission areas of the domain, in particular in the Bodélé depression in Chad, the Chech desert in Algeria and the El Djouf between Mauritania and Mali, followed by the Arabian desert, the Taklamakan desert in Northwest China and the smaller areas of the Grand Erg Oriental in Algeria, the Grand Sand Sea between Lybia and Egypt, and the Kharan desert in south-western Pakistan.

Table 3 reports the averaged DOD of first-guess, analysis and analysis minus first-guess (analysis increments) when calculated for the whole domain for the full period, for different seasons (DJF, MAM, JJA, SON), and for individual years.

Earth System
Open Access Science
Data Discussions

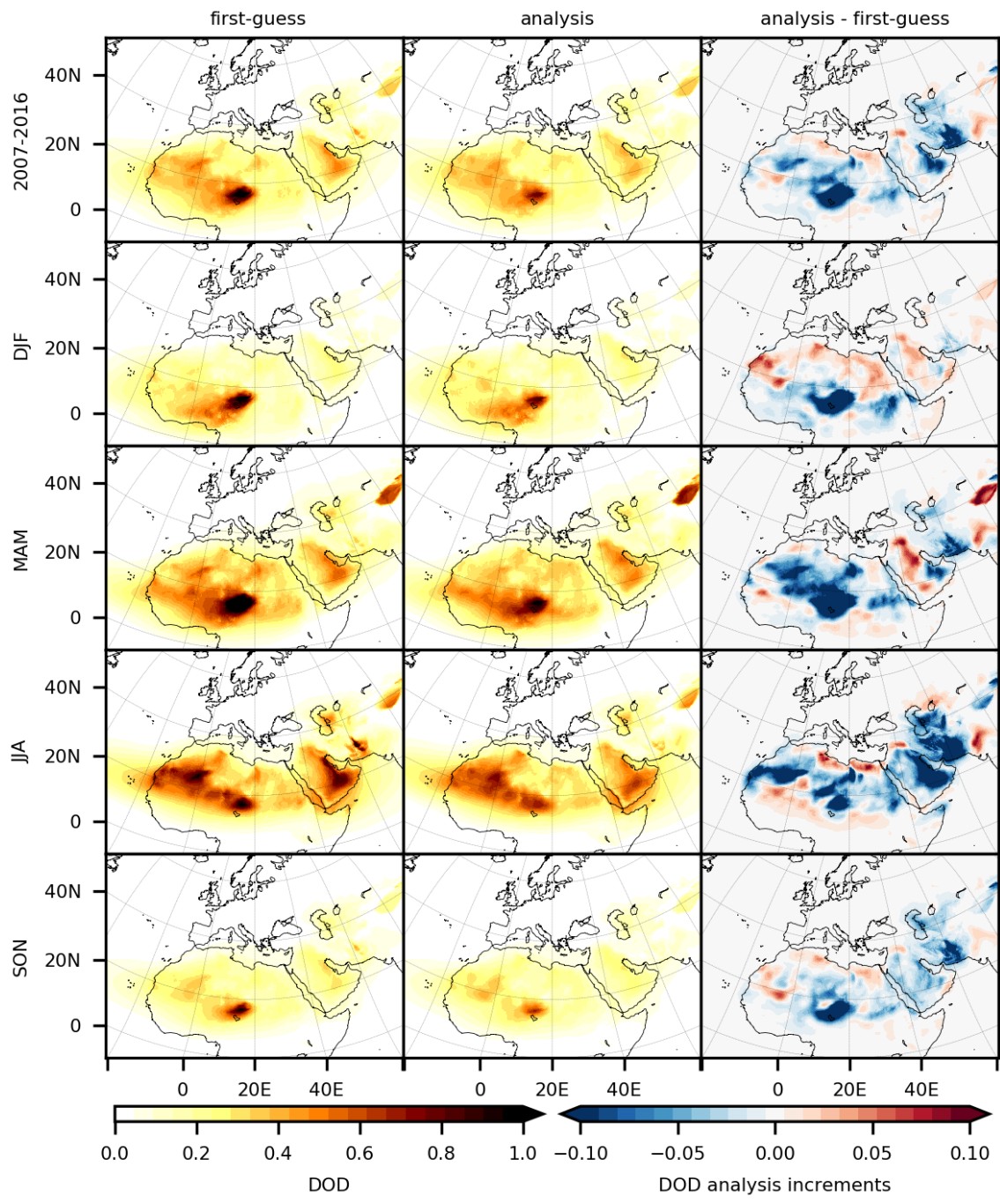

**Figure 6.** Maps of mean 3-hourly DOD first-guess (first column), analysis (second column), and analysis increments (third column) calculated for the whole period (2007-2016; top row) and for different seasons (DJF, MAM, JJA, SON; rows 2 to 5). Model fields are ensemble mean.





**Table 3.** Averaged DOD of first-guess (fg), analysis (an) and analysis increments (an-fg) for the full period (2007-2016), for different seasons (DJF, MAM, JJA, SON), and for individual years.

| period | mean fg DOD | mean an DOD | mean analysis increments |
|---|---|---|---|
| 2007-2016 | 0.1066 | 0.1 | -0.0066 |
| DJF | 0.0806 | 0.0781 | -0.0025 |
| MAM | 0.1353 | 0.1268 | -0.0084 |
| JJA | 0.1364 | 0.1261 | -0.0103 |
| SON | 0.0734 | 0.0681 | -0.0053 |
| 2007 | 0.107 | 0.0994 | -0.0076 |
| 2008 | 0.1185 | 0.1117 | -0.0068 |
| 2009 | 0.1049 | 0.0993 | -0.0056 |
| 2010 | 0.1091 | 0.1041 | -0.005 |
| 2011 | 0.1056 | 0.0994 | -0.0063 |
| 2012 | 0.1113 | 0.1055 | -0.0058 |
| 2013 | 0.0986 | 0.091 | -0.0076 |
| 2014 | 0.0943 | 0.0879 | -0.0064 |
| 2015 | 0.1125 | 0.1046 | -0.0079 |
| 2016 | 0.1044 | 0.0968 | -0.0075 |

The decadal mean analysis DOD (top row of Fig. 6) is generally smaller than the first-guess DOD except in the Taklamakan and Thar deserts, and in areas where the mean DOD is below 0.3. Therefore on average, MONARCH emissions are likely too strong for the configurations used, although a potentially too weak deposition cannot be discarded. The latter is strongly
dependent upon the emitted size distribution that evolves during transport.

Seasonal changes in the geographical distribution of the analysis mean DOD (rows 2 to 5 of Fig. 6) are consistent with well-known patterns (Prospero et al., 2002; Ginoux et al., 2012): (i) dust peaks everywhere during spring and summer, in particular, across the Taklamakan desert during spring when more dust-generating cold fronts arrive in the area; (ii) dust from the south Sahara and Sahel is preferentially transported by northeasterly Harmattan winds toward the Gulf of Guinea in winter
and spring; (iii) the dust plume originated in Western Africa and transported across the tropical North Atlantic is shifted toward northern latitudes in summer along with the Intertropical Convergence Zone (ITCZ; Moulin et al., 1997); (iv) dust is strongly mobilized in the Arabian peninsula and the Tigris-Euphrates basin in summer by the north-northwesterly Shamal winds; (v) the lowest overall DOD is simulated everywhere in autumn.

### 8.2 DOD analysis increments

Figure 6 also shows the difference between DOD analysis and first-guess (namely analysis increments; right column) averaged over the full reanalysis period (top row). Non-zero systematic analysis increments are to be interpreted as systematic corrections





on the model simulations and can serve as a proxy for model bias. By applying these corrections, the analysis improves the underlying model. The patterns of these systematic corrections vary with season and geographical location. While over the entire domain the mean analysis and first-guess are comparable, the biggest systematic negative corrections (removing mass
from the atmosphere) are linked to overestimation of sources' strengths in the Bodélé depression in Chad, in the Saudi Arabia lowlands and in the Balochistan region of south-western Asia that extends over Iran, Afghanistan and Pakistan, and contains, for example, the Kharan desert. Negative mean increments are also present but to a lesser extent in other arid and semi-arid areas such as the Erg Chech desert in Algeria, the Great Sand Sea in Libya, the Nubian desert in Sudan and eastwards of the Caspian Sea. Positive mean increments calculated for the whole reanalysis period are less widespread than the negative increments.
The strongest values are over the Thar desert, in the northern part of Syria, a long stretch inland from the Mediterranean Sea in the north of Africa, and in the desert of El Djouf between Mauritania and Mali. All in all, as expected, the largest positive or negative analysis increments correspond to areas with more dust load, i.e., to source regions and their vicinity.

The patterns of the mean increments depend upon the season (see row 2 to 5 of Fig. 6). These patterns are clearly linked to the seasonal changes in dust activities in the different regions, as mean increments are, in absolute value, higher in the presence
of high mean DODs compared to low DOD values. The areas that show the strongest seasonality with respect to the analysis increments are the Bodélé depression, and the Arabian and Taklamakan deserts. The overestimation of the Bodélé source strength in the first-guess is more pronounced in winter and spring. In spring the emissions from the Taklamakan desert, Syria and the northern part of the Arabian desert are clearly underestimated, while in summer strong negative increments are present all over the Arabian desert. Wide areas in the Sahara desert are affected by negative increments in the spring and summer.
The Balochistan region and the Thar desert show, respectively, negative and positive increments throughout the year, but their magnitudes are greater in spring and summer.

The patterns of the increments are consistent among the different years (Fig. S2 and S3) and vary mostly in the amplitude of the mean corrections, although there are some exceptions. Positive increments over the Thar desert, northern Syria and the north of the Arabian desert mainly appear in the first part of the reanalysis, between 2007 and 2012, in contrast to the small
positive or even negative increments in the case of the Arabian desert in the subsequent years. Strong negative increments east of the Caspian Sea are applied mainly through 2007 to 2010. Those yearly differences suggest changes, for example in land use, that are not captured by the model. Negative corrections in the west of the Sahara are more widespread in 2007 and 2008 than in other years due to the higher mean DOD during those two years.

### 8.3 Statistics of departures from assimilated observations

We compare here the reanalysis $DOD_{coarse}$ with the assimilated observations. Figure 7 shows the $DOD_{coarse}$ for the observation-collocated ensemble mean first-guess and analysis, and for the assimilated observations averaged over the full reanalysis period (top row of Fig. 7) and over the DJF, MAM, JJA, and SON seasons (from the second to the fifth row of Fig. 7). Table 4 reports the corresponding values averaged over the whole domain for the full period, for different seasons, and for individual years. By visual inspection, the analysis is closer to the assimilated observations in all the time periods considered, which constitutes
a good sanity check for the assimilation scheme. This is confirmed also when the averages are calculated for individual years





of the reanalysis period (Fig. S4 and S5). The seasonality in the model simulations closely resembles that in the observations, with MAM and JJA being the most active dust seasons.

**Table 4.** Averaged $DOD_{coarse}$ of observation-collocated first-guess (fg), observation-collocated analysis (an) and assimilated MODIS DB retrievals for the full period (2007-2016), for different seasons (DJF, MAM, JJA, SON), and for individual years.

| period | mean fg $DOD_{coarse}$ | mean an $DOD_{coarse}$ | mean MODIS DB $DOD_{coarse}$ |
|---|---|---|---|
| 2007-2016 | 0.1914 | 0.1685 | 0.1912 |
| DJF | 0.1445 | 0.1374 | 0.1573 |
| MAM | 0.2323 | 0.2074 | 0.2337 |
| JJA | 0.2452 | 0.2073 | 0.2356 |
| SON | 0.1427 | 0.1228 | 0.1394 |
| 2007 | 0.1905 | 0.1644 | 0.1858 |
| 2008 | 0.2108 | 0.1877 | 0.2114 |
| 2009 | 0.1892 | 0.1682 | 0.1921 |
| 2010 | 0.1947 | 0.1772 | 0.202 |
| 2011 | 0.1894 | 0.1679 | 0.192 |
| 2012 | 0.195 | 0.1752 | 0.1989 |
| 2013 | 0.1832 | 0.1577 | 0.1771 |
| 2014 | 0.1768 | 0.1546 | 0.176 |
| 2015 | 0.1948 | 0.1697 | 0.1929 |
| 2016 | 0.1896 | 0.1621 | 0.1836 |

Figure 8 shows the mean (first and second column of Fig. 8) and standard deviation (third and fourth column of Fig. 8) of the first-guess and analysis $DOD_{coarse}$ departures (respectively) from assimilated observations averaged over the full reanalysis period (top row of Fig. 8) and over the different four seasons (from the second to the fifth row of Fig. 8). The corresponding values averaged over the whole domain are reported in Table 5, together with the number of observation counts and statistics calculated for individual years. The departure statistics, and in particular the reduction of the standard deviation of the analysis departures compared to the first-guess everywhere in the domain of interest, prove the consistency of our assimilation procedure. This is the case also on a seasonal and yearly basis (Fig. S6 and S7). With respect to the mean departures, the positive mean departures (model simulation minus observations) decrease considerably in the analysis compared to the first-guess, while some of the negative mean departures remain unchanged in specific regions or seasons. The latter is the case for example in Europe and Russia, when considering the full reanalysis period or the different seasons. The aforementioned regions see on average much lower $DOD_{coarse}$ values than the rest of the domain and are analyzed less efficiently. This is likely due to the ensemble not having a sufficient spread for low simulated concentrations. A similar issue was previously identified in other assimilation systems (see Benedetti et al., 2009, Sect. 4) and attributed to the fact that aerosol mass is a positive definite variable, which intrinsically deviates from the assumed Gaussian conditions in the prior in the analysis step. Negative mean

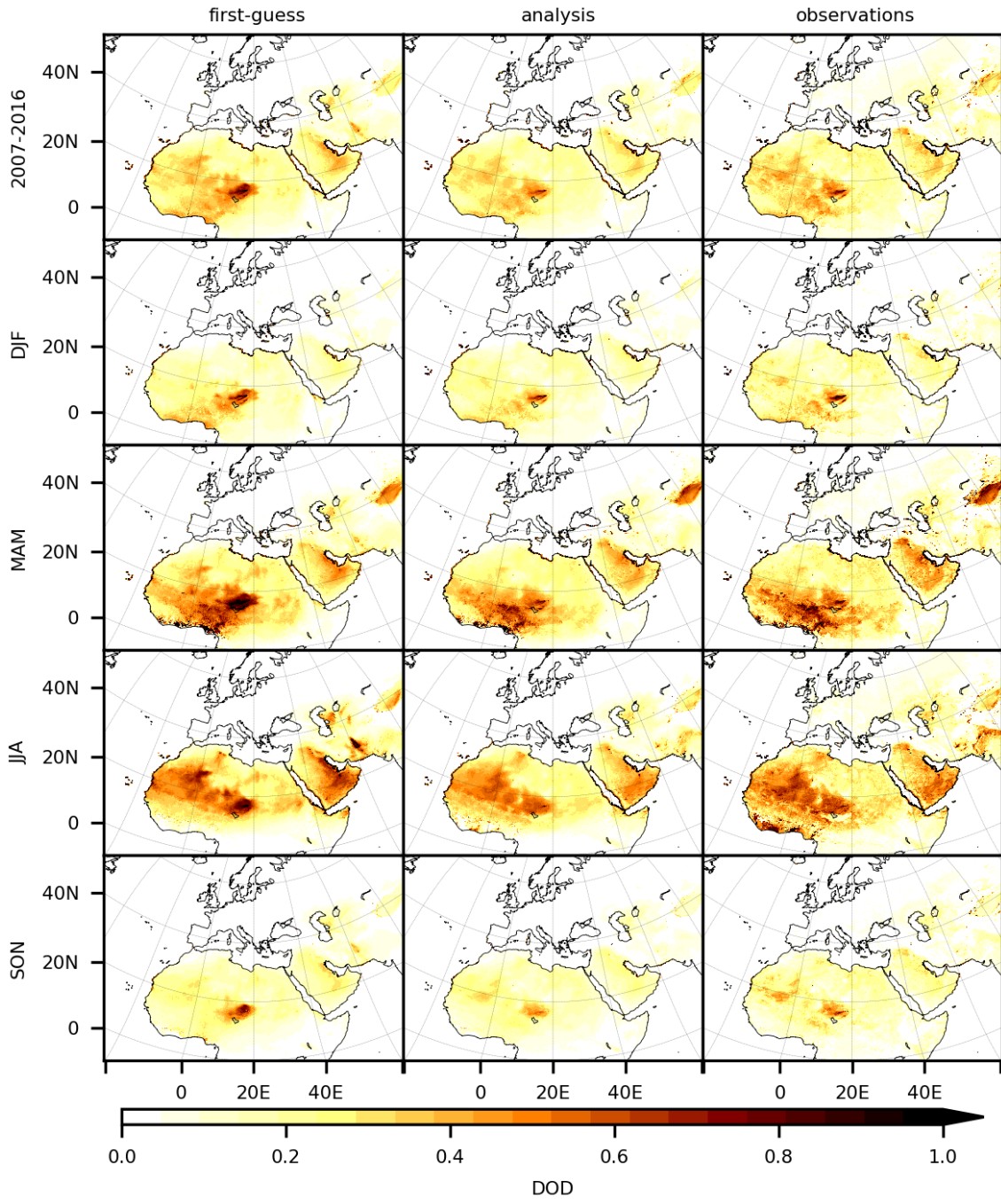

**Figure 7.** Maps of mean 3-hourly $DOD_{coarse}$ first-guess (first column), analysis (second column) and MODIS DB assimilated observations (third column) calculated for the whole period (2007-2016; top row) and for different seasons (DJF, MAM, JJA, SON; rows 2 to 5). Model fields are ensemble mean and are collocated with the observations.





departures are present in the sub-Sahel region, and in particular over a stretch along the Gulf of Guinea in summer season, both with respect to the first-guess and the analysis. This might be due to the contamination of other aerosols than dust in the observational data set, which might be of anthropogenic or natural origin, e.g. aerosol produced by biomass burning in Central

Africa advected northward (Haslett et al., 2019). Due to the above regional or seasonal issues, the total bias (i.e. mean departure from the observations calculated over the entire domain) is higher in the analysis compared to the first-guess.

**Figure 8.** Maps of mean 3-hourly $DOD_{coarse}$ for first-guess departures (first column), analysis departures (second column), standard deviation of first-guess departures (third column) and standard deviation of analysis departures (fourth column) calculated for the whole period (2007-2016; top row) and for different seasons (DJF, MAM, JJA, SON; rows 2 to 5). Model fields are ensemble mean and are collocated with the observations.





**Table 5.** Statistics (mean and std) of departures of $DOD_{coarse}$ first-guess (fg) and analysis (an) from assimilated observations calculated for the full period (2007-2016), for different seasons (DJF, MAM, JJA, SON), and for individual years. Number of observation counts are also reported.

| period | observation counts | mean of fg departures | mean of an departures | std of fg departures | std of an departures |
|---|---|---|---|---|---|
| 2007-2016 | $4.373 \times 10^8$ | 0.0002 | -0.0227 | 0.1841 | 0.0985 |
| DJF | $1.008 \times 10^8$ | -0.0128 | -0.0198 | 0.1502 | 0.0827 |
| MAM | $1.033 \times 10^8$ | -0.0014 | -0.0262 | 0.212 | 0.1125 |
| JJA | $1.157 \times 10^8$ | 0.0096 | -0.0283 | 0.2159 | 0.112 |
| SON | $1.176 \times 10^8$ | 0.0033 | -0.0166 | 0.1357 | 0.0744 |
| 2007 | $4.522 \times 10^7$ | 0.0047 | -0.0214 | 0.1863 | 0.0961 |
| 2008 | $4.398 \times 10^7$ | -0.0006 | -0.0237 | 0.1928 | 0.1023 |
| 2009 | $4.274 \times 10^7$ | -0.0029 | -0.0239 | 0.1858 | 0.0984 |
| 2010 | $4.464 \times 10^7$ | -0.0073 | -0.0248 | 0.1828 | 0.1011 |
| 2011 | $4.43 \times 10^7$ | -0.0026 | -0.0240 | 0.1825 | 0.1008 |
| 2012 | $4.417 \times 10^7$ | -0.0039 | -0.0237 | 0.1913 | 0.1554 |
| 2013 | $4.299 \times 10^7$ | 0.006 | -0.0194 | 0.1767 | 0.0969 |
| 2014 | $4.41 \times 10^7$ | 0.0008 | -00214 | 0.1762 | 0.0924 |
| 2015 | $4.348 \times 10^7$ | 0.0019 | -0.0232 | 0.1924 | 0.1047 |
| 2016 | $4.17 \times 10^7$ | 0.0059 | -0.0215 | 0.1823 | 0.0952 |

## 8.4 Verification of DOD and $DOD_{coarse}$ against AERONET

We compare here the reanalysis DOD and $DOD_{coarse}$ with independent observations that have not been used in the assimilation process. We employed products from the Aerosol Robotic Network (AERONET) of ground-based Sun photometers (Holben et al., 1998; O'Neill et al., 2003; Giles et al., 2019).

### 8.4.1 Verification methodology

We used AERONET version 3 quality-assured data. On the one side, the modelled $DOD_{coarse}$ at 550 nm is compared with coarse-mode AOD retrievals at 500 nm from the spectral de-convolution algorithm (SDA; O'Neill et al., 2003). In general, $AOD_{coarse}$ is dominated by maritime/oceanic aerosols and desert dust. However, sea-salt is usually associated to low AOD (< 0.03; Dubovik et al., 2002) and mainly affects coastal stations, and therefore inland high $AOD_{coarse}$ values can be assumed to be mineral dust. On the other side, the modelled DOD at 550 nm is compared with dust-filtered AOD values from the direct-sun algorithm (Giles et al., 2019). We used direct-sun AOD retrievals between 440 and 870 nm to obtain the AOD at 550 nm using the Ångström law. Dust-dominated conditions are identified using a specific set of dust filters based on the AERONET





Ångström exponent (AE). AE is inversely related to the average size of the particles: the smaller the particles, the larger AE. AE
ranges normally from 4, corresponding to pure molecular extinction, down to close to null values, corresponding to extinction
dominated by coarse-mode aerosols (sea-salt and mineral dust) producing a spectral neutral AOD (O'Neill et al., 2003). Values
of AE > 1.2 typically indicate significant presence of fine-mode particles (biomass burning or urban aerosols; Basart et al.,
2009). Quantitative evaluations of the modelled DOD are conducted for dust-dominated conditions based on four different AE
filters (Table 6) where AE ranges from desert dust source typical values (AE < 0.4) to values characteristic of dust long-range
transport conditions (AE < 0.75). Additionally, for one of the filter methods (namely DOD-mixed2), DOD is assumed to be 0
when the AE is greater than 1.2. These dust-filters roughly represent "pure" desert dust conditions (i.e. DOD-dust1, DOD-dust2
and $DOD_{coarse}$) and long-range transport (i.e. mixed) dust conditions (i.e. DOD-mixed1 and DOD-mixed2).

**Table 6.** Dust filters applied to the AERONET retrievals.

| Filter name | Dust condition | Dust presence | Non-dust presence |
|---|---|---|---|
| DOD-dust1 | Pure dust | DOD = AOD when AE<0.40 | - |
| DOD-dust2 | Pure dust | DOD = AOD when AE<0.60 | - |
| DOD-mixed1 | Mixed dust | DOD = AOD when AE<0.75 | - |
| DOD-mixed2 | Mixed dust | DOD = AOD when AE<0.75 | DOD=0 when AE>1.2 |
| $DOD_{coarse}$ | Pure dust | $DOD_{coarse} = AOD_{coarse}$ | - |

We focus our verification on the NAMEE region. We used data from 139 AERONET stations (Fig. 9, see also Table S3 to
S8 for the list of used AERONET sites), which includes all the available AERONET sites in the NAMEE domain providing
observations during the period of 2007-2016, with the exception of those sites that are at high-altitudes (i.e. altitudes greater
than about 1800 m above sea level). Results are presented for different sub-regions, namely the Middle East, Northwestern
Africa and the Mediterranean, and for all available AERONET stations including those sites outside the three above-mentioned
regions, and that are depicted in Fig. 9. Model values are ensemble mean analysis fields.

AERONET measurements are nominally taken at 15-minute intervals. Here we average observations within ±30 minutes of
the 3-hourly model output times. These averaged observations are used to evaluate the model on 3-hourly, daily and monthly
basis. For the daily and monthly average evaluation, only coincident 3-hourly model output and observations are used. We use
verification statistics such as the Pearson correlation coefficient (r), mean bias (MB), root mean square error (RMSE) and mean
fractional bias (MFB) (see Appendix B) to measure the skill of the model when performing diagnostic analyses of DOD and
$DOD_{coarse}$ where AERONET sites are located.

**8.4.2  Comparison with 3-hourly, daily and monthly reference data**

Overall, the dust reanalysis can reproduce the 3-hourly, daily and monthly observed variability with Pearson correlation coef-
ficients ranging from 0.74 and 0.82, depending on the dust filter, for 3-hourly DOD to up to 0.9 for monthly $DOD_{coarse}$. The
reanalysis tends to underestimate the DOD and $DOD_{coarse}$ compared to AERONET observations (see Fig. 10). The model re-



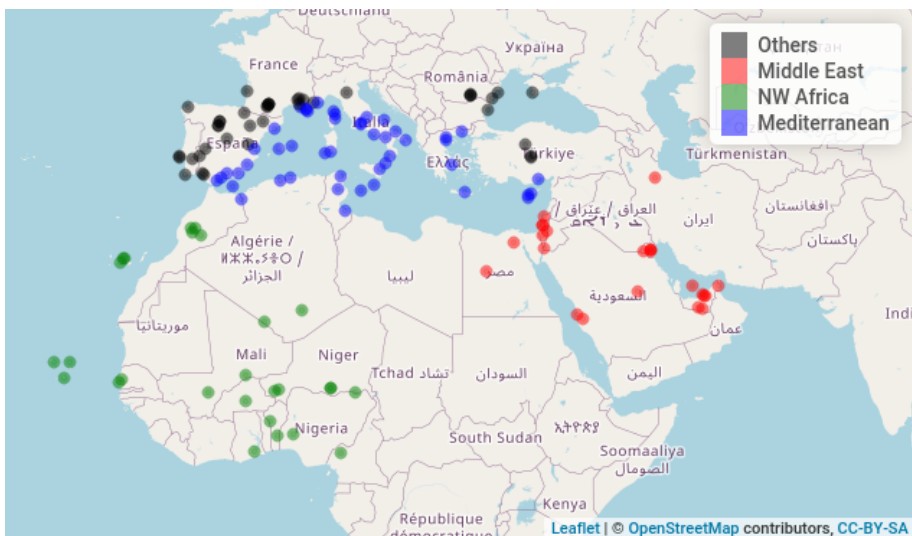

**Figure 9.** Spatial distribution of the AERONET sites used in this study. The different colours indicate the sub-regions considered in the discussion of the results: Northwestern Africa (green), the Middle East (red) and the Mediterranean (blue). When validating the NAMEE region, all the sites are considered, including the ones labelled Others (black).

sults are dominated by the results in Northwestern Africa and the largest relative underestimations are observed in the Mediter-

ranean and the Middle East (Fig. 11), likely because of marine aerosols at these sites. Therefore some model underestimation is expected in particular in the proximity of coastal stations or when mixtures of aerosols are present (Basart et al., 2009).

Table 7 to 9 present the verification statistics on a 3-hourly, daily and monthly basis when calculated using the five dust-filtered reference data sets. The stricter the dust filter, the lower the correlation coefficient.

The verification results calculated using the DOD-mixed2 dust filter are comparable to those obtained with the $DOD_{coarse}$

reference data set in terms of correlation (0.82 versus 0.81 for the entire region), and MB (-0.04 versus -0.05). When considering regional results, the use of the DOD-mixed2 dust filter shows a reduction of the MB together with an increase of MFB in the Mediterranean region (Fig. 12). This is directly related to the assumption DOD = 0 for AE > 1.2 (Table 6), which increases the number of collocations particularly in the Mediterranean, where the presence of dust is sporadic. This is confirmed by the comparison with the results obtained with the DOD-mixed1 filter where this condition is neglected (see Fig. 12). The

RMSE obtained with DOD-mixed2 and $DOD_{coarse}$ reference data shows a clear north to south gradient that scales with dust concentrations with maximum values over sources (in Northwestern Africa and the Middle East, RMSE > 0.12) and minimum values in the Mediterranean (RMSE < 0.12).

Monthly DOD and $DOD_{coarse}$ verification statistics are sensitive to the number of AERONET observations, as shown in Fig. 13. A clear seasonal trend is identified with lower performance in the cloudy winter season than in summer when clear

skies are more frequent. Time series of the verification statistics for $DOD_{coarse}$ show a change after 2011, with reductions in MB and RMSE in comparison to previous years. Also the MFB is closer to the MFB from the different dust filters (see Fig.

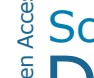

**Table 7.** Verification statistics (r, RMSE, MB and MFB) and number of samples (NDATA) for the reanalysis versus AERONET AODs for the entire period (2007-2016) and NAMEE region, and for Northwestern Africa, the Middle East and Mediterranean regions. AERONET version 3, cloud-screened, 3-hourly, dust-filtered AOD and $AOD_{coarse}$ is the reference. The definition of each of the DOD filters is in Table 6.

|  | DOD-dust1 | DOD-dust2 | DOD-mixed1 | DOD-mixed2 | $DOD_{coarse}$ |
|---|---|---|---|---|---|
| NAMEE region |  |  |  |  |  |
| NDATA | 68209 | 99418 | 121645 | 259277 | 288944 |
| r | 0.74 | 0.75 | 0.76 | 0.82 | 0.81 |
| RMSE | 0.25 | 0.22 | 0.21 | 0.15 | 0.12 |
| MB | -0.10 | -0.10 | -0.10 | -0.04 | -0.05 |
| MFB | 0.42 | -0.51 | -0.58 | 0.79 | -1.00 |
| Northwestern Africa |  |  |  |  |  |
| NDATA | 8997 | 16999 | 23775 | 88261 | 102672 |
| r | 0.57 | 0.61 | 0.64 | 0.75 | 0.69 |
| RMSE | 0.22 | 0.18 | 0.17 | 0.09 | 0.08 |
| MB | -0.12 | -0.11 | -0.10 | -0.02 | -0.05 |
| MFB | -0.68 | -0.79 | -0.89 | 1.22 | -1.26 |
| Middle East |  |  |  |  |  |
| NDATA | 15278 | 23655 | 29819 | 41115 | 51256 |
| r | 0.66 | 0.67 | 0.68 | 0.72 | 0.72 |
| RMSE | 0.24 | 0.22 | 0.21 | 0.19 | 0.15 |
| MB | -0.10 | -0.09 | -0.09 | -0.04 | -0.03 |
| MFB | -0.27 | -0.29 | -0.30 | 0.33 | -0.15 |
| Mediterranean Basin |  |  |  |  |  |
| NDATA | 40377 | 51494 | 57101 | 59614 | 51556 |
| r | 0.76 | 0.77 | 0.77 | 0.78 | 0.79 |
| RMSE | 0.26 | 0.24 | 0.24 | 0.23 | 0.19 |
| MB | -0.10 | -0.10 | -0.10 | -0.09 | -0.08 |
| MFB | -0.35 | -0.40 | -0.43 | -0.33 | -0.51 |

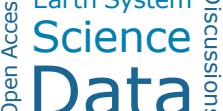

**Table 8.** Same as Table 7 but using daily reanalysis values and AERONET dust-filtered AOD and AOD$_{coarse}$ averaged on a daily basis as reference.

| | DOD-dust1 | DOD-dust2 | DOD-mixed1 | DOD-mixed2 | DOD$_{coarse}$ |
|---|---|---|---|---|---|
| **NAMEE region** | | | | | |
| NDATA | 28553 | 40618 | 49366 | 49366 | 103540 |
| r | 0.77 | 0.79 | 0.79 | 0.79 | 0.84 |
| RMSE | 0.23 | 0.21 | 0.20 | 0.20 | 0.11 |
| MB | -0.11 | -0.10 | -0.10 | -0.10 | -0.05 |
| MFB | -0.50 | -0.59 | -0.67 | -0.67 | -1.02 |
| **Northwestern Africa** | | | | | |
| NDATA | 4467 | 7967 | 10920 | 10920.00 | 36764 |
| r | 0.61 | 0.65 | 0.67 | 0.67 | 0.72 |
| RMSE | 0.20 | 0.17 | 0.16 | 0.16 | 0.08 |
| MB | -0.12 | -0.11 | -0.10 | -0.10 | -0.05 |
| MFB | -0.79 | -0.91 | -1.01 | -1.01 | -1.27 |
| **Middle East** | | | | | |
| NDATA | 5929 | 8820 | 10862 | 10862 | 17470 |
| r | 0.70 | 0.72 | 0.72 | 0.72 | 0.77 |
| RMSE | 0.22 | 0.20 | 0.19 | 0.19 | 0.13 |
| MB | -0.11 | -0.10 | -0.09 | -0.09 | -0.04 |
| MFB | -0.28 | -0.29 | -0.31 | -0.31 | -0.17 |
| **Mediterranean Basin** | | | | | |
| NDATA | 16058 | 19787 | 21681 | 21681 | 18880 |
| r | 0.78 | 0.79 | 0.79 | 0.79 | 0.81 |
| RMSE | 0.25 | 0.24 | 0.23 | 0.23 | 0.19 |
| MB | -0.10 | -0.10 | -0.10 | -0.10 | -0.08 |
| MFB | -0.37 | -0.42 | -0.46 | -0.46 | -0.54 |



**Table 9.** Same as Table 7 but using monthly reanalysis values and AERONET dust-filtered AOD and AOD$_{coarse}$ averaged on a monthly basis as reference.

| | DOD-dust1 | DOD-dust2 | DOD-mixed1 | DOD-mixed2 | DOD$_{coarse}$ |
|---|---|---|---|---|---|
| NAMEE region | | | | | |
| NDATA | 1189 | 1227 | 1239 | 1239 | 960 |
| r | 0.81 | 0.84 | 0.85 | 0.85 | 0.90 |
| RMSE | 0.19 | 0.17 | 0.16 | 0.16 | 0.12 |
| MB | -0.11 | -0.11 | -0.10 | -0.10 | -0.08 |
| MFB | -0.39 | -0.41 | -0.43 | -0.43 | -0.45 |
| Northwestern Africa | | | | | |
| NDATA | 1298 | 1716 | 1908 | 1908 | 1998 |
| r | 0.62 | 0.62 | 0.67 | 0.67 | 0.76 |
| RMSE | 0.19 | 0.16 | 0.14 | 0.14 | 0.05 |
| MB | -0.13 | -0.12 | -0.11 | -0.11 | -0.04 |
| MFB | -0.90 | -0.96 | -1.01 | -1.01 | -1.05 |
| Middle East | | | | | |
| NDATA | 691 | 798 | 837 | 837 | 795 |
| r | 0.73 | 0.72 | 0.75 | 0.75 | 0.85 |
| RMSE | 0.17 | 0.15 | 0.13 | 0.13 | 0.09 |
| MB | -0.10 | -0.09 | -0.08 | -0.08 | -0.04 |
| MFB | -0.30 | -0.32 | -0.33 | -0.33 | -0.15 |
| Mediterranean Basin | | | | | |
| NDATA | 1189 | 1227 | 1239 | 1239 | 960 |
| r | 0.81 | 0.84 | 0.85 | 0.85 | 0.90 |
| RMSE | 0.19 | 0.17 | 0.16 | 0.16 | 0.12 |
| MB | -0.11 | -0.11 | -0.10 | -0.10 | -0.08 |
| MFB | -0.39 | -0.41 | -0.43 | -0.43 | -0.45 |

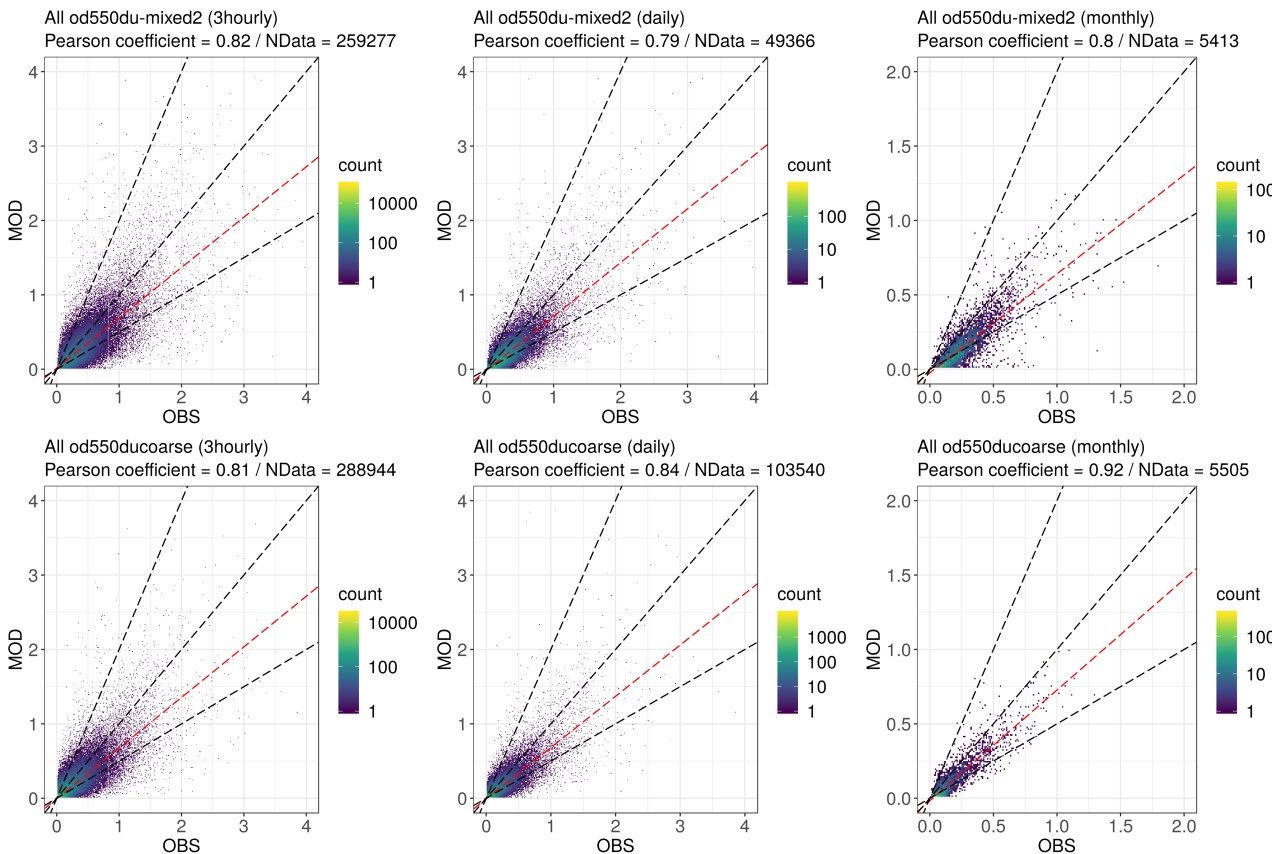

**Figure 10.** Density scatter plots of the reanalysis (MOD) DOD and DOD$_{coarse}$ versus AERONET (OBS) dust-filtered AOD, DOD-mixed2 (top row) and DOD$_{coarse}$ (bottom row), during the whole reanalysis period (2007-2016). The results are calculated for the NAMEE domain and for different time basis: 3-hourly (left), daily (middle) and monthly (right). The dust filters applied to the AERONET observations are described in Table 6. Bin size is 0.01.

13). This change is associated to a decrease in DOD$_{coarse}$ in the Mediterranean region (not shown here) that is captured by the reanalysis. Underestimations are observed in northwestern Africa and the Mediterranean regions when the DOD-mixed2 and DOD$_{coarse}$ data are used as reference. In summertime in Northwestern Africa, we find the largest underestimations (monthly

MB < -0.10 for DOD-mixed2 and DOD$_{coarse}$). These underestimations are likely related to strong dust outbreaks associated with mesoscale convective systems (called Haboobs) that the model is not able to capture. In the Middle East, the model shows a systematic underestimation when compared to DOD-mixed2 and DOD$_{coarse}$ reference data, although some overestimation in particular years (2011-2012) is observed. The observed DOD underestimations in comparison with AERONET in the Middle East can be partly attributed to a poor representation of small-scale emission processes such as the wind peak associated with

the breakdown of the nocturnal low-level jet, the meteorological effects in the vicinity of complex topography, sea breezes and cold pools (Basart et al., 2016).



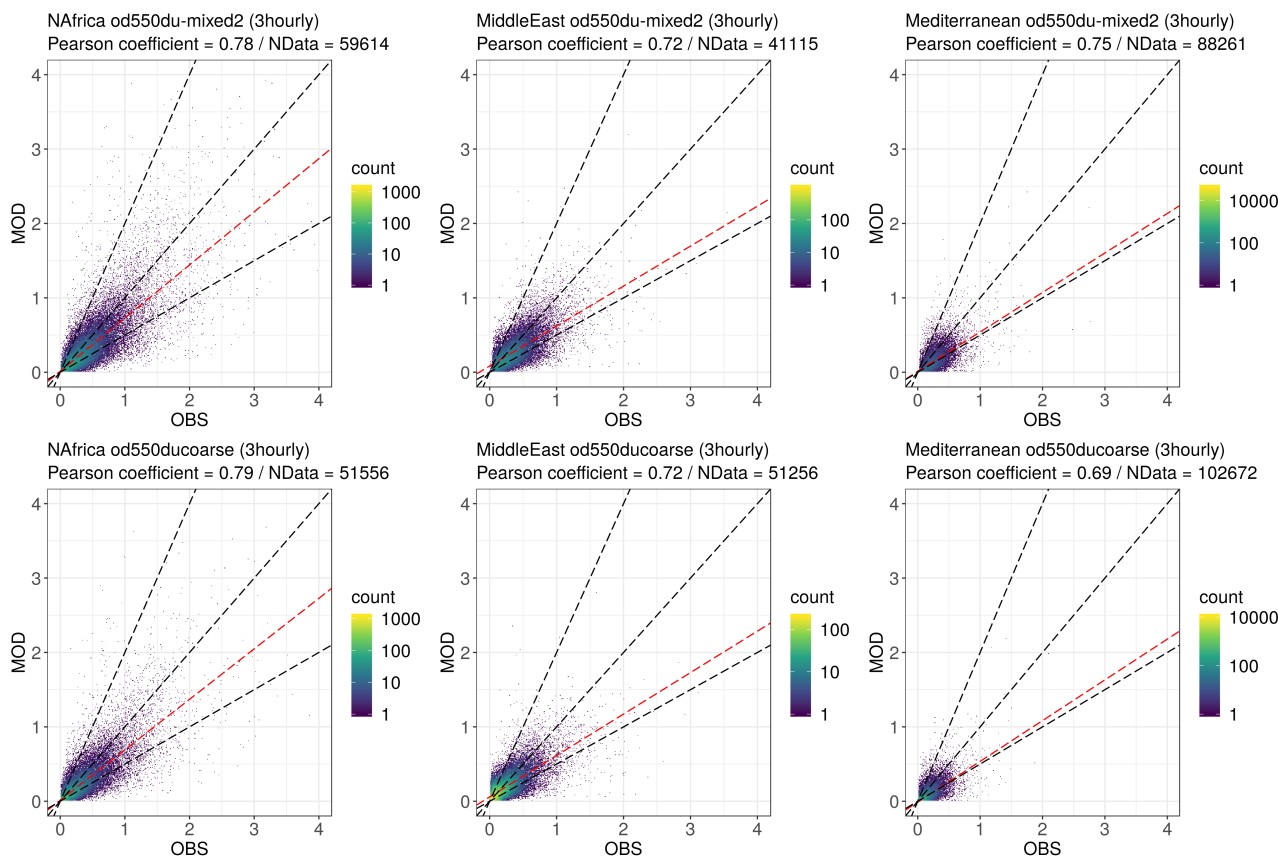

**Figure 11.** Density scatter plots of the reanalysis (MOD) DOD and DOD$_{coarse}$ versus AERONET (OBS) dust-filtered AOD, DOD-mixed2 (top row) and DOD$_{coarse}$ (bottom row), during the whole reanalysis period (2007-2016) and on a 3-hourly basis. The results are calculated for 3 different sub-regions of the reanalysis domain: Northwestern Africa (first column), the Middle East (second column) and the Mediterranean (third column). The dust filters applied to the AERONET observations are described in Table 6. Bin size is 0.01.

Overall, the comparison with AERONET observations shows a good performance of the reanalysis in reproducing the spatial and temporal distribution of mineral dust aerosols over the entire domain and for the 10-year period.

## 9 Data availability

The reanalysis data set (Di Tomaso et al., 2021) is distributed via a Thematic Real-time Environmental Distributed Data Service (THREDDS) at BSC and made freely available at http://hdl.handle.net/21.12146/c6d4a608-5de3-47f6-a004-67cb1d498d98 (to request access during the review process, please contact reanalysis.access@bsc.es). Additionally, for review purposes a password-protected, anonymous access has been set up. The data set (78 TB in size) is structured into individual NetCDF files per geophysical variable and type of ensemble statistics (ensemble mean, standard deviation, maximum and median). Each

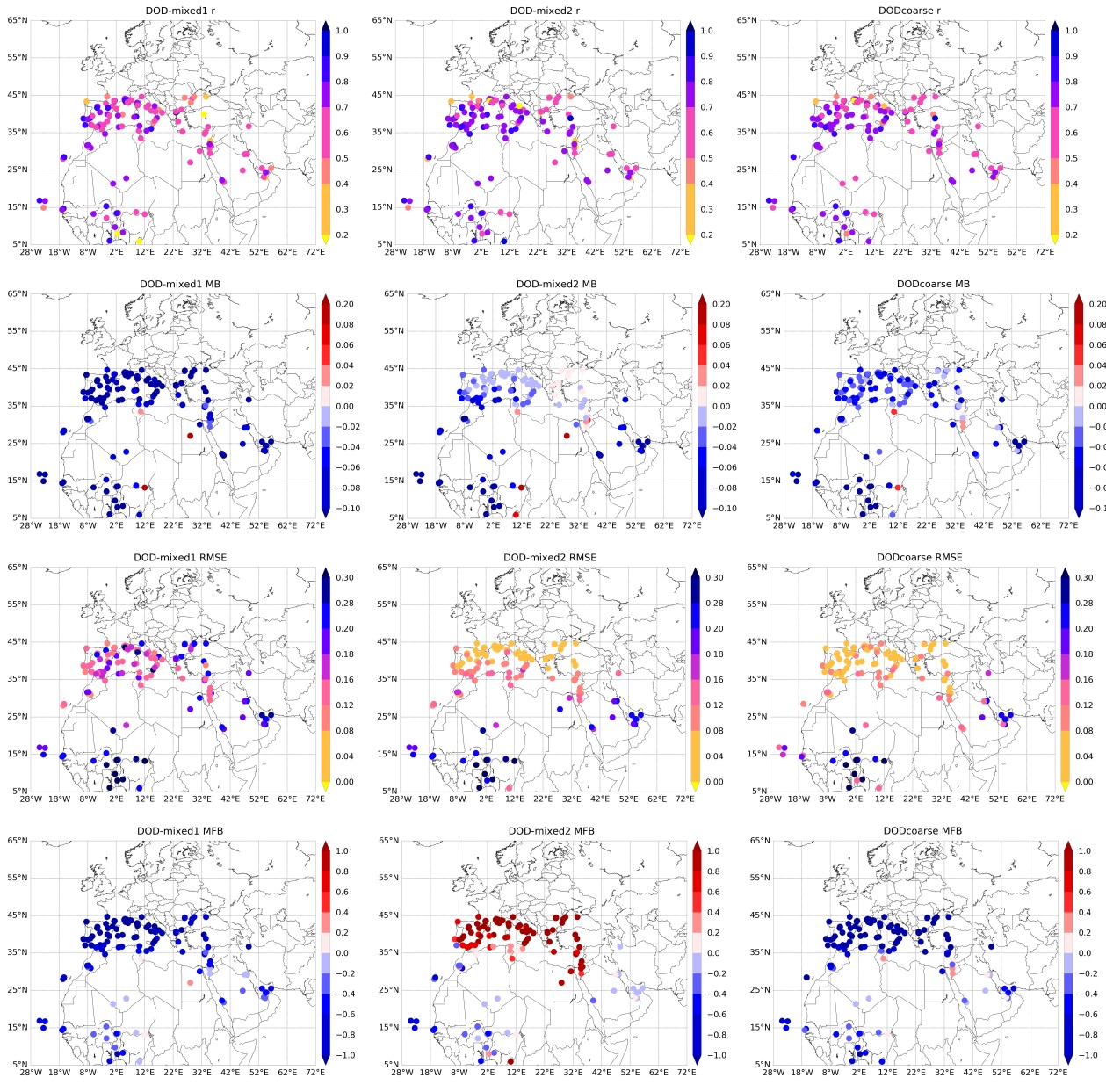

**Figure 12.** Maps of verification statistics (r, MB, RMSE, MFB, from top to bottom) of the analysis DOD (first and second columns) and DOD$_{coarse}$ (third column) versus AERONET dust-filtered AOD (Table 6): DOD-mixed1 (left), DOD-mixed2 (middle) and DOD$_{coarse}$ (right) calculated for the whole period (2007-2016). The results are obtained using 3-hourly collocated reanalysis and observation values (see also Table 7).



**Figure 13.** Time series of monthly verification statistics (r, MB, RMSE, MFB) and number of samples (NDATA) for the reanalysis DOD and DOD$_{coarse}$ versus dust-filtered AERONET observations for the period 2007-2016 for the NAMEE domain. Different colours are associated to the results obtained with the different dust-filters: DOD-dust1, DOD-dust2, DOD-mixed1, DOD-mixed2 and DOD$_{coarse}$. The definitions of each dust-filter is reported in Table 6. The results are obtained using 3-hourly collocated model and observation values.



individual file covers the time period of one assimilation window (24 hours) and contains 8 time steps at a 3-hourly time-frequency starting at 3 UTC. The files are organized into folders, where each folder contains the files relative to the whole reanalysis period (10 years) for a given variable and type of statistics.

## 10 Conclusions and further perspectives

A regional dust reanalysis has been produced using the MONARCH chemical weather prediction system and satellite retrievals
of $DOD_{coarse}$ based on MODIS-Aqua DB AOD at 550 nm. The reanalysis data set spans the period 2007-2016 at a horizontal resolution of $0.1°$ latitude $\times$ $0.1°$ longitude, and a temporal resolution of 3 hours. The reanalysis covers a regional domain centered around northern Africa, the Middle East and Europe (NAMEE region), that also includes parts of Central Asia, and the Atlantic and Indian Oceans. This paper describes the modelling, observational and assimilation aspects related with the production of the reanalysis, whose unprecedented high resolution has required the use of advanced archiving and computing
strategies, which are also described in the paper (see Appendix C). The assimilated observations have provided a total column optical constraint on the coarse fraction of dust particles over land, in cloud- and snow-free conditions, and at day-time, with one satellite overpass per day. Analysis increments were estimated over the whole assimilation window through the 4D implementation of the assimilation algorithm and, to a certain extent (limited by the observation radius of influence), also over sea according to the model background spatial covariance. Re-partitions of analysis increments in the vertical dimension of the
model control vector and across the individual model coarse size bins have relied on the model background.

 The seasonal changes of the spatial distribution of the dust reanalysis are well characterized and follow well-known, region-dependent dust cycle features controlled by seasonal changes in meteorology (mainly surface winds but also precipitation) and in vegetation cover. The most prominent seasonal features that stand out in the reanalysis are the mobilization of dust during the so-called Asian dust events in the Taklamakan region in spring, and by north-northwesterly Shamal winds in the Arabian
peninsula and Tigris-Euphrates basin during summer, the transport of south Saharan dust south-west toward the Gulf of Guinea by northeasterly Harmattan trade winds during winter and spring, and the northward shift of the plume extending from Western Africa over the tropical Atlantic during summer due to movements of the ITCZ.

 Diagnostics based on departures of first-guess and analysis from assimilated observations provided a sanity check for the quality of our assimilation procedure. As expected, the analysis is statistically closer to the assimilated observations than the
first-guess. The mean departures are larger in the analysis than in the first-guess only in specific regions and seasons, which can be explained by the contamination of aerosols other than dust in the observational data set (for example, biomass burning aerosols produced by fires in Central Africa and that are advected further north during summer) or by the presence of fairly low $DOD_{coarse}$ values (mainly over Europe and west Russia) that are not analyzed as efficiently by the assimilation scheme as the higher $DOD_{coarse}$ values. Mean analysis increments suggest seasonally-dependent model biases that follow dust seasonal
changes. By applying these corrections, the analysis improves the underlying model. Overall, the spatial distribution of the analysis increments over source regions, and in their proximity, highlights the pivotal role of the MODIS DB retrievals in



providing an observational constraint over the most critical regions, confirming what previous studies have shown (Di Tomaso et al., 2017; Benedetti et al., 2019).

The reanalysis DOD and DOD$_{coarse}$ have been validated with highly accurate, ground-truth measurements from AERONET on a 3-hourly, daily and monthly basis and with the application of specific dust filters on the reference products or the use of the coarse-mode AOD product. When the latter is used as reference, a Pearson correlation coefficient as high as 0.81 with a MB of -0.05 and a RMSE of 0.12 are estimated when considering the whole reanalysis period and 3-hourly AERONET retrievals. This confirms the good accuracy of the reanalysis data set, and its suitability to be used in specific air quality/health and climate service applications. By extending the existing observational-based information intended for mineral dust monitoring, this reanalysis will allow a better quantification of dust impacts upon key sectors of society and economy. This makes the data set a potentially useful tool in support of climate research and service, including the support to operational early warning systems and to the development of mitigation strategies.

This desert dust reanalysis data set is intended to be the first major endeavor towards the production of BSC aerosol reanalyses over regional or global domains. Extensions of the data set are planned for the near future. A series of companion papers will provide a more comprehensive evaluation of the reanalysis, an analysis of inter-annual variability and trends, and a description of its application in dust-tailored services.

**Appendix A: Folder and file naming convention of the reanalysis data set**

As described in Sec. 9, the reanalysis data set is structured into individual files per variable and type of ensemble statistics (e.g., an individual file contains the ensemble mean analysis of DOD at 550 nm). The filenames include the following terms separated by the underscore sign: the short name of the variable (as reported in Table 2), the initial date and time of the data included in the file, a suffix among av, max, median, std indicating, respectively, the ensemble mean, max, median, standard deviation for that variable, and optionally the label an for the variables for which an analysis field is produced. When the latter label is not present, the fields are model first-guess. The filenames end with the extension suffix nc identifying NetCDF files. Each individual file contains 8 time steps at a 3-hourly time-frequency starting at 3 UTC. Therefore, for example, the file name for the ensemble mean analysis of DOD at 550 nm for a given date is od550du_YYYYMMDDHH_av_an.nc, where od550du is the variable short name and YYYYMMDDHH can take values from 2007010103 to 2016123103. The files are organized into folders containing the whole 10-year period for a given variable and type of statistics. Each folder is named with the variable short name followed by the hyphen sign and the suffix indicating the type of ensemble statistics, and optionally by the label an preceded by the underscore sign, for the variables for which there is an analysis field. Hence the folder containing the files of the example above is named od550du-av_an, while the corresponding ensemble mean first-guess are stored in the folder named od550du-av. To follow on with the same example, the ensemble mean analysis of DOD at 550 nm for July 9, 2012 can be found in the file path od550du-av_an/od550du_2012070903_av_an.nc.



## Appendix B:  Verification metrics

The definition of the verification metrics used in this study are reported in Table B1.

**Table B1.** Definitions of the verification statistics used in the study. $o_i$ and $c_i$ are the observed and the modelled concentrations at time and location $i$, respectively, $\bar{o}$ and $\bar{c}$ are their averages, and $n$ is the number of data.

| Statistic Parameter | Formula |
|---|---|
| Pearson correlation coefficient ($r$) | $r = \dfrac{\sum\limits_{i=1}^{n} (c_i - \bar{c}) \cdot (o_i - \bar{o})}{\sqrt{\sum\limits_{i=1}^{n} (c_i - \bar{c})^2} \cdot \sqrt{\sum\limits_{i=1}^{n} (o_i - \bar{o})^2}}$ |
| Mean bias (MB) | $MB = \dfrac{1}{n} \sum\limits_{i=1}^{n} (c_i - o_i)$ |
| Root mean square error (RMSE) | $RMSE = \sqrt{\dfrac{1}{n} \sum\limits_{i=1}^{n} (c_i - o_i)^2}$ |
| Mean fractional bias (MFB) | $MFB = \dfrac{2}{n} \sum\limits_{i=1}^{n} \left( \dfrac{c_i - o_i}{c_i + o_i} \right)$ |

## Appendix C:  Simulation workflow

The reanalysis has been run on the BSC High Performance Computing (HPC) infrastructure using the Autosubmit workflow manager (Manubens et al., 2016; Uruchi et al., 2021), a Python-based tool to create, manage and monitor experiments running in one or multiple remote computing clusters or HPC via Secure Shell protocol. Scripts and templates to use Autosubmit were developed specifically for the reanalysis to be able to easily run and monitor long simulations by using the BSC HPC
resources and store their results in the BSC archive. Autosubmit handles the job submission of the different workflow steps automatically, taking into account interruptions and failures. A functionality to wrap the 12 daily model simulations (each using 768 computing cores) and the data assimilation calculations (using 576 cores) was used to minimize the queuing times. This allows processing a number of days in a row and to increase the parallelism since a single job allocates the total sum of computing nodes required (i.e. 9792 cores were reserved by each wrapper). The number of computing cores for each job
has been estimated to balance the execution time of a model simulation and of the assimilation for these jobs to share the computing resources. Additionally, the jobs of two post-processing steps were also wrapped together with the simulation and data assimilation jobs. The post-processing steps are needed to compress and reduce the original model and assimilation output, and for the calculation of some basic ensemble statistics (to be provided to the users as main output of the reanalysis), before the final output is transferred to a long-term archive. The job wrapper has been designed with a crossing-date strategy to run
two different starting dates within the same experiment. This is done so that the model ensemble simulations from the first starting date can be run in parallel with the assimilation job of the second starting date. This choice of design was made since model ensemble simulations and data assimilation calculations from the same date cannot be run simultaneously, due to the



obvious dependency of the data assimilation job on the model output. This novel workflow design was developed specifically for the production of the dust reanalysis and has proven successful for such highly computationally-intensive calculations.

*Author contributions.* CPGP, SB, EDT, OJ, JE designed the study and discussed the main results. FB, EC, PF, LM, MM, AV, EW, MK, MG contributed to the discussions. EDT, JE, FM, NS developed and maintained the data assimilation code. PG prepared the assimilated observations. OJ leads the MONARCH developments with contributions from CPGP, MG, MK, VO, JE, EDT. SB and CPGP carried out the independent validation, while EDT and JE performed the validation against assimilated observations. FM, MC, GM, EDT, MO contributed to the development of the computing workflow. FM, AB, GM, EDT run the simulations. AB, SB, EDT, PA, FB performed the data set quality

check and storage. EDT, CPGP, SB wrote the manuscript. All authors commented on the manuscript.

*Competing interests.* The authors declare that they have no competing interests.

*Acknowledgements.* We acknowledge the DustClim project which is part of ERA4CS, an ERA-NET initiated by JPI Climate, and funded by FORMAS (SE), DLR (DE), BMWFW (AT), IFD (DK), MINECO (ES), ANR (FR) with co-funding by the European Union's Horizon 2020 research and innovation programme (Grant n. 690462). BSC co-authors also acknowledge support from the European Research Council

under the European Union's Horizon 2020 research and innovation programme (grant n. 773051; FRAGMENT), the AXA Research Fund, the Spanish Ministry of Science, Innovation and Universities (grant n. RYC-2015-18690 and CGL2017-88911-R), the European Union's Horizon 2020 research and innovation programme (grant n. 792103; SOLWARIS). This work has been partially funded by the contribution agreement between AEMET and BSC to carry out development and improvement activities of the products and services supplied by the WMO Sand and Dust Storm Regional Centres. Jerónimo Escribano and Martina Klose have received funding from the European Union's Horizon 2020

research and innovation programme, respectively, under the Marie Skłodowska-Curie grant agreements H2020-MSCA-COFUND-2016-754433 and H2020-MSCA-IF-2017-789630. Martina Klose further acknowledges support through the Helmholtz Association's Initiative and Networking Fund (grant agreement n. VH-NG-1533). We acknowledge PRACE (eDUST, eFRAGMENT1, and eFRAGMENT2) and RES (AECT-2019-3-0001, AECT-2020-1-0007, AECT-2020-3-0013) for awarding access to MareNostrum at the BSC and for providing technical support. The authors thank all the Principal Investigators and their staff for establishing and maintaining the NASA and PHOTONS

AERONET sites, and the MODIS mission scientists and associated NASA personnel for the production of the data used in this study. We would like to thank John P. Dunne who provided constructive comments that improved the manuscript.



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
