# Peer review of "The MONARCH high-resolution reanalysis of desert dust aerosol over Northern Africa, the Middle East and Europe (2007-2016)"

_Earth System Science Data, 2021_

## Author Comment (AC1)

Dear Editor, dear Reviewers,
we would like to thank you for considering our paper for publication in ESSD. We have appreciated the constructive comments of the reviewers, and we have addressed all of them through the answers below and modifications in the original submitted manuscript.

Additionally, we would like to note that we have applied slight changes in the validation study against AERONET data due to a refinement and correction of the selection criteria for the AERONET stations. All AERONET sites with altitude < 1850 m above sea level have been considered, while in the submitted version some sites above this altitude threshold (e.g. one site in Morocco at 2400 m above sea level) had been erroneously included. We have specified this in line 478 of the revised manuscript. Furthermore, the number of observation samples for DODcoarse has been revised and corrected. We have updated Tables 7-9, S3-S8 and Figures 9-13 accordingly. The overall conclusions remain valid and unchanged compared to the submitted version.

We have also updated some of the cited references that are in preparation or have been published during the review process. We have also corrected some typos: removed "in prep." when citing papers in preparation. We have added the DOI, when missing, and corrected the chronological order of some references. We have added the date of last access when citing websites. We have corrected the year of publication of a reference (Schroedter-Homscheidt et al., 2013). We have corrected a co-author surname (adding the second surname Pinto to Gilbert Montané). We have described better one of the variables (height of pressure level above sea level). We have specified that the reanalysis horizontal resolution refers to a rotated latitude longitude grid. As already specified in the submitted manuscript (now line 122 of the revised manuscript), MONARCH's "regional version used in this work runs on a rotated latitude-longitude grid."

We have updated the "Data availability" section since the details for accessing the data set are now available at the PID provided.

**Authors' response to reviewer #1**

We thank the reviewer for their positive comments about our work and about the usefulness of this manuscript for potential users of the data set. We wish to thank the reviewer also for the constructive comments which helped us to improve the clarity of the manuscript.

In the following we respond to the specific comments. Reviewer's comments are repeated in *italic*.

*1) The manuscript describes a validation of the data set in terms of AOD (or specific, DOD, Dust Optical Depth). This is also the quantity that is assimilated, and it therefore makes sense to use this as first validation. For a data set related to dust, it would however be useful to have also an idea on the dust concentrations themselves, and how accurate these are. The only dust-related results are shown in Figure 4, but no comparison with observations has been made. Will there be a validation of the dust concentrations included in the follow-up papers mentioned in Section 7? It would be useful to have that clearly mentioned. Also, some remarks could be made already on the dust concentrations themselves and how they are changed by the analysis.*

The two papers that are cited in Section 7 of the submitted manuscript are dealing with the validation of dust extinction coefficient, either total integrated column or vertical profiles. Indeed, dust concentrations will be evaluated in an additional follow-up paper for air quality applications in Europe, whose reference has been added to the revised version of the manuscript in the lines 351 and 374. We have also added a comment about that work in lines 350-352 of Section 7 of the revised manuscript.

*2) For example, what is the impact of the calibration described in Section 6.1 on the dust load in the ensemble members? If my interpretation is correct, the calibration factors for the dust emissions range from 0.004 for the K14 emission scheme, to 2.65 for the MB95 scheme. This is a huge difference; does it mean that the K14 scheme by default has a huge over-estimation? After calibration, do the ensemble members have dust concentrations that are more or less in the same range?*

The aim of the calibration of our simulations is to reduce the overall systematic bias of the ensemble and to have all the members in the same range of values, as the reviewer has correctly stated, since simulations are calibrated against the same reference values. The calibration of dust emission fluxes is a common and necessary practice since dust emission processes occur at smaller temporal and spatial scales than those resolved by regional and global models. For example, model winds tend to be underestimated over sources and the bare surface erodible fraction is very uncertain and likely overestimated. Additionally, emission estimates are non linear with respect to wind speed, and a calibration is required in particular to adjust those estimates to be consistent with the model spatial resolution, meteorology, and the soil property databases used in the simulation. The K14 scheme does indeed use a very small calibration factor, also in other models, because scaling up field measurements can indeed cause a large overestimation of the fractional area that is actually emitting dust (Jasper Kok, personal communication, 2 November 2015).

*3) The adjustment of the dust concentrations depends strongly on how DOD is calculated, thus on the optical properties and the radiance computations. Is there any idea on how accurate these computations are? With incorrect optical properties computed, the dust concentrations might require unrealistic perturbations to obtain the correct DOD's. The meteorological data is also relevant for this computation I guess; since this comes in the ensemble from two different models (MERRA2 and ERA-Interim), is there a clear difference seen between the DOD's computed for different meteo?*

Regarding the optical properties, we acknowledge that they are a source of uncertainty in atmospheric dust modelling (e.g., Kok et al., 2017). Assumptions on particle shape, size and refractive indices have an effect on DOD, and this has a potential impact on dust concentration both in the calibration and assimilation steps. We agree that a follow-up study on the impact of using different assumptions for the characteristics of dust in our system would be interesting to pursue, while for this description paper we find that such a study is beyond the scope of the journal. Regarding the meteorological data, the use of different meteorological initial and boundary conditions introduces additional perturbations in our ensemble compared to source perturbation only, and aims at representing model uncertainty in the different processes where the meteorology has a role (transport, deposition and emission). This is in line with the studies by Escribano et al. (2021, 2022) which show the benefit of using an ensemble of meteorological initial and boundary conditions in our system, as well as with the study by Rubin et al. (2016) which shows the benefit of combining meteorological and aerosol source perturbation in aerosol data assimilation. We have added further clarifications about this point made by the reviewer in lines 184-185 as well as an additional reference in lines 187-188 of the revised manuscript.

*4) Some clarification on the ensemble generation would be useful. Section 3 describes that a 12-member ensemble is used, with each member choosing one-of-two meteo sets, one-of-three emission schemes, and a random value for (among others) the friction threshold; is that indeed what is done? I guess that each member then keeps it's choice for meteo and emission scheme, but are the emission parameters changing in time or per grid cell?*

Yes, as we explained in lines 194-198 of the revised manuscript each ensemble member is run with one of the three emission schemes, one of the two meteorological initial and boundary conditions, and with different perturbed parameters for the emission scheme. The perturbation parameters don't change in time and per grid, as in Di Tomaso et al. (2017). We have clarified this point in lines 205-206 of the revised manuscript.

*5) Lines 145-146: What does a Frequency-of-Occurrence of 0.20 mean? That in 20% of the days dust is observed over a location?*

Not exactly: the 0.2 value refers to DOD, while the threshold that we have used for the Frequency-of-Occurrence (FoO) is 0.05. As stated in lines 145-147 of the revised manuscript, "The location of dust sources is identified by a climatology of frequency of occurrence (FoO) of DOD greater than 0.2 [ . . . ] with a minimal threshold for FoO equal to 0.05, below which there is no emission." Therefore, dust can be emitted if the retrieved FoO(DOD > 0.2) is greater than 0.05, that is 5% of the days a value of DOD>0.2 is observed. Please note that this is a necessary but not sufficient condition as other requirements have to be satisfied for dust to be emitted in the model.

*6) Line 230: What is meant with a time slot centered around 12 UTC? Aren't more MODIS orbits assimilated then, with different time slots?*

MODIS observations are assimilated at 12 UTC, and thanks to the use of a 4D extension of the LETKF they affect the whole assimilation window. We have clarified this point in lines 232-233 of the revised manuscript. Please also note that, given the meridional extent of the simulation domain, 12 UTC is a reasonable approximation for the 3-hourly model collocation with Aqua 1:30 p.m. equatorial crossing local time.

*7) The analysis weights in Eq. (2) might provide some information on a preference of the analysis for certain ensemble members, for example the members with a specific emission scheme. Is that indeed possible, and has some information been deduced from them?*

We don't store the weight matrix since it is an internal step of the data assimilation algorithm and it would require large storage resources for this model resolution and time span of the reanalysis. The ensemble is used to provide a description of prior uncertainty as a key ingredient of the data assimilation algorithm, rather than an evaluation of the different possible configurations of the MONARCH model. Emissions are one of the biggest sources of uncertainty, hence the use of different schemes has allowed us to represent part of this uncertainty. We appreciate the suggestion of the reviewer for a study aiming to improve dust modelling. In this respect, Klose el al. (2021) describes the validation of some of the different schemes that are implemented in MONARCH, showing that they have different merits and weaknesses according to the aspects considered.

*8) Section 6 describes that an observation screening is applied. Is it kept which fraction of the observations has been rejected, and whether that is especially in certain regions?*

We have not stored such information. Removing outliers (observations that largely depart from the model prior) is a normal procedure in data assimilation. We appreciate the suggestion of the reviewer for a study aiming to assess the observational dataset from a model perspective, which we consider is out of the scope of this description paper.

*9) Table 6: What is exactly done for "DOD-mixed2"? How could AE be <0.75 and >1.2 ?*
*Line 492: The "DOD-mixed2" leads to more zero values; should that be visible in Figure 10 then as a higher density?*

The dust filter "DOD-mixed2" assigns DOD = AOD when AE<0.75, while assigns DOD=0 when AE>1.2. Therefore, with respect to AE, it is either AE<0.75 or AE>1.2. The explanation for the design of this filter is indicated in the manuscript: AOD observations associated with "AE > 1.2 typically indicate significant presence of fine-mode particles (biomass burning or urban aerosols; Basart et al.,2009). Quantitative evaluations of the modelled DOD are conducted for dust-dominated conditions based on four different AE filters (Table 6) where AE ranges from desert dust source typical values (AE < 0.4) to values characteristic of dust long-range transport conditions (AE < 0.75)."

This criteria produces indeed more zero values in the dust reference data set than when using the other filters. This increased number of zero values is shown in Table 7 as indicated by an increased number of observations in DOD-mixed2 with respect to DOD-mixed1 (that only accounts for observations with AE < 0.75). This is also visible in the number of observations (NDATA) of Figure 13 (top panel): the DOD-mixed2 data set has more reference samples than when using the DOD-mixed1 filter. Please, note that Figure 10 shows the comparison between DOD-mixed2 and DODcoarse that corresponds to different AERONET database retrievals.

*10) Line 502. Is the change in statistics associated with changed conditions? Or could it be related to a degradation of the data?*

As we say in line 510 of the revised manuscript, "This change is associated with a decrease in DODcoarse in the Mediterranean region", hence it is a change in the conditions.

*11) Line 510: Which complexity of the topography is relevant here?*

We are mentioning here the meteorological effects in the vicinity of mountains. As it is shown in Basart et al. (2016), "The complex orography in south-western Saudi Arabia, Yemen and Oman (with peaks higher than 3000 m) has an impact on the transported dust concentration fields over mountain regions. Differences between both model configurations [i.e. 30km and 3km horizontal resolution] are mainly associated with the channelization of the dust flow through valleys and the differences in the modelled altitude of the mountains that alters the meteorology and blocks the dust fronts limiting the dust transport." We have changed the sentence by using the "meteorological effects of orography" in line 518 of the revised manuscript.

*12) Line 78: chemical formula should not in math mode*

We have corrected this. Thank you.

*13) Line 79: "additionally"*

We have corrected this typo in the paper. Thank you.

*14) Lines 197, 199: shouldn't it be: 'friction velocity "for" wind velocity' ?*

Depending on the emission scheme, a friction velocity or a wind velocity is used (Klose et al, 2021, Sect 3.1.1). To clarify this point we have modified the sentence in lines 198-199 of the revised manuscript specifying that for one of the schemes the threshold wind velocity is perturbed.

*15) Subsection 6.1: shouldn't this be a section?*

We prefer to leave 6.1 as a subsection of Section 6 since it provides information about simulation settings.

**Authors' response to reviewer #2**

We wish to thank the reviewer for the positive comments on the document, the content of the data set and of its metadata.

In the following we respond to the specific comments. Reviewer's comments are repeated in *italic*.

*1) Some confusion are made on between Modis Collection 6 and collection 6.1 (some clarifications are needed).*

We have assimilated DOD retrievals from MODIS Collection 6 as stated in line 208 and in the caption of Figure 1 of the revised manuscript. Other collections of MODIS products that are mentioned in the manuscript refer either to what the JRAero reanalysis has assimilated or the MODIS Collection 5 monthly Leaf Area Index used with the MB95 emission scheme.

*2) In the table 1 which is the overview of the experiment that has generated the data is missing the data assimilation window (it is mentioned at the earliest in section 5, adding it in the table will be beneficial).*

We have added in Table 1 the length of the data assimilation window. Thank you for the suggestion.

*3) The dust bins description is referred but again a table making an easy finding of the information will be helpful for someone who would like to use the data.*

We have added the details about the particle size bins in Table 1. Thank you for the suggestion.

*4) The organisation of the sections should be revisited (the section 6 should come directly after section 3 as it is more details on the model.*

We wish not to change the order of the sections since section 6 will not be sufficiently clear if the basic theory about data assimilation, as well as the details about the observations, are not explained first. Section 6 in fact provides the details of the specific settings that we have used for the aspects described in Sections 2 to 5. We have clarified further this point in lines 303-304 of the revised manuscript with a sentence at the start of Section 6 specifying that this section describes the details of the settings for the modelling, observational and data assimilation aspects described in the previous four sections.

*5) The output of this experiment should be compared to a denial experiment where no data assimilation will be performed to understand why the emission scheme is over producing (issue with the scheme and the climatology behind? scheme loaded with dust transport? ...) That will be interesting to conduct maybe for an other article.*

Klose et al. (2021) describes in detail the emission schemes implemented in MONARCH, as well as the validation results for four different schemes. They show that each scheme has merits and limitations according to the different aspects considered. We appreciate the suggestion of the reviewer for another work to take this study further, while for this description paper we find that such a study is beyond the scope of the journal. We wish to add that some results on the comparison with a denial experiment, where we show that for a period of 2012 the analysis outperforms a simulation without data assimilation, are in a paper presented at the 38[th] ITM conference and are currently under consideration by Springer to be included as a chapter in the volume entitled "Air Pollution Modeling and its Application XXVIII". Due to copyright issues we cannot reproduce the

plots in this answer to the reviewer, but we have added its reference in the line 374 of the revised manuscript.

*6) The comparison with an independent set of data: Modis is evaluated against AERONET, so if your system is converging toward your data by your data assimilation, it seems logical that it will also converge toward AERONET even if it is not a guarantee. I would suggest that you add another independant det of data in the evaluation process for the eventual article that I have suggested before (lidar, dry deposition measurement, ...).*

Further validation and comparison of the reanalysis dataset with independent observational data sets will be described in follow-up papers: Mytilinaios et al. (2022a, 2022b) and Barnaba et al. (2022a). Following the reviewer's comment, we have added a sentence in lines 374-375 of the revised manuscript specifying further details on the reference data sets used in the follow-up studies.